# EPIC: Effective Prompting for Imbalanced-Class Data Synthesis in Tabular Data Classification via Large Language Models

**Jinhee Kim**[*]
KAIST
Daejeon, South Korea
seharanul17@kaist.ac.kr

**Taesung Kim**[*]
KAIST
Daejeon, South Korea
zkm1989@kaist.ac.kr

**Jaegul Choo**
KAIST
Daejeon, South Korea
jchoo@kaist.ac.kr

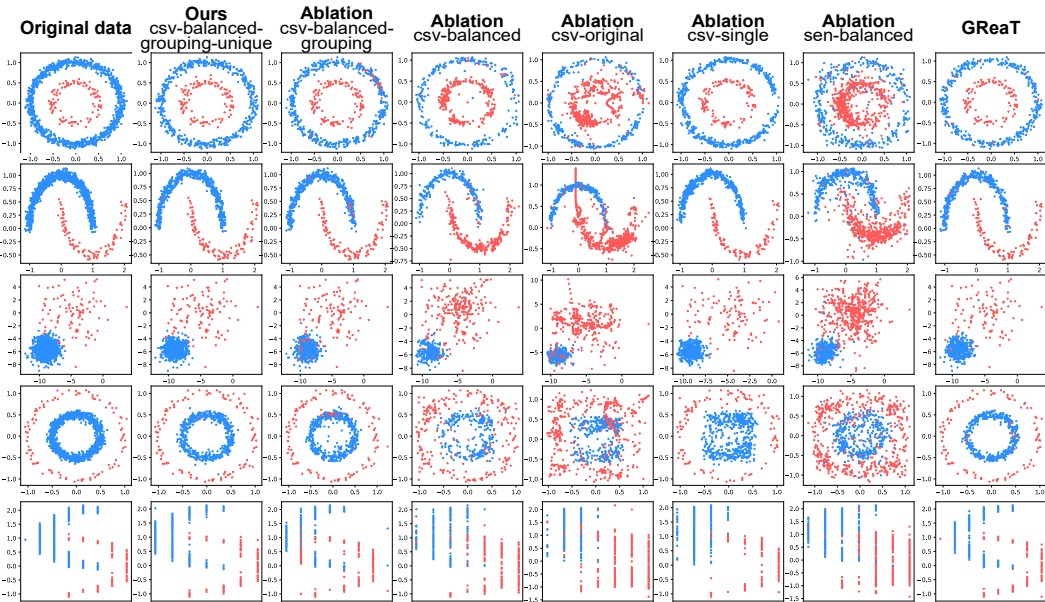

Figure 1: **Generation results on an imbalanced toy dataset with majority and minority classes**. Our approach, leveraging in-context learning with LLMs, achieves (1) distinct class boundaries, (2) accurate feature correlations, (3) well-matched value ranges, (4) robust numerical-categorical relationships (last row), and (5) comprehensive data distribution coverage, with improvements over its ablated versions and the fine-tuned GReaT model [4]. Complete results are available in Fig. 9.

## Abstract

Large language models (LLMs) have demonstrated remarkable in-context learning capabilities across diverse applications. In this work, we explore the effectiveness of LLMs for generating realistic synthetic tabular data, identifying key prompt design elements to optimize performance. We introduce **EPIC**, a novel approach that leverages balanced, grouped data samples and consistent formatting with unique variable mapping to guide LLMs in generating accurate synthetic data across all classes, even for imbalanced datasets. Evaluations on real-world datasets show that EPIC achieves state-of-the-art machine learning classification performance, significantly improving generation efficiency. These findings highlight the effectiveness of EPIC for synthetic tabular data generation, particularly in addressing class imbalance. Our source code for our work is available ***here***.

---

[*]Both authors contributed equally.

38th Conference on Neural Information Processing Systems (NeurIPS 2024).

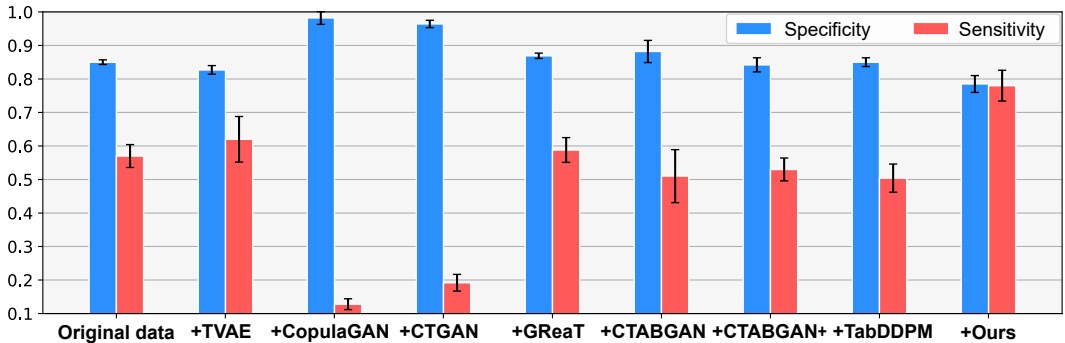

Figure 2: **Comparison of ML classification performance with synthetic data on the Travel dataset.** Results are averaged across four classifiers: XGBoost, CatBoost, LightGBM, and gradient boosting classifier, with each classifier run five times. Our method uses the GPT-3.5-turbo model.

# 1  Introduction

Tabular data, consisting of mixed variable types such as numerical and categorical variables, represents a widely applicable and essential data format [30]. It plays a crucial role in enhancing decision-making and efficiency in various real-world applications, such as finance, healthcare, manufacturing, and natural sciences [1, 3, 26]. However, challenges such as data scarcity and class imbalance, particularly for rare yet crucial events, often significantly degrade the performance of machine learning (ML) models, often resulting in reduced accuracy for underrepresented classes. To address these challenges, efficient synthetic data generation methods have been developed to augment tabular datasets. Traditional methods, such as synthetic minority oversampling technique (SMOTE) and its variants [5, 11, 21], focus on generating minority class samples to alleviate class imbalance. Recently, advanced generative models, such as TVAE [35], CTAB-GAN [39], and TabDDPM [19], have shown promising results in producing high-quality synthetic tabular data.

Large language models (LLMs) have also demonstrated significant potential for generating realistic tabular data, effectively handling both numerical and categorical variables [4, 37]. However, these models often require extensive, data-specific fine-tuning, which can increase the risk of overfitting to majority classes or dominant feature values, especially in small and highly imbalanced tabular datasets. An alternative approach involves leveraging the in-context learning capabilities of LLMs. Existing work has shown that LLMs can solve complex reasoning tasks, learn and mimic patterns from input data, and augment textual datasets without parameter updates [13, 20, 34].

However, designing effective prompts to optimize this capability for tabular data generation is challenging, especially for imbalanced datasets. This is because tabular data is not naturally expressed in textual form and involves unique challenges, such as identifying feature correlations and accurately representing underrepresented attributes. Therefore, carefully crafted prompt design tailored to synthetic tabular data generation is essential to fully leverage the capabilities of LLMs. While several studies have employed in-context learning with LLMs for synthetic tabular data generation [27, 37], few have conducted comprehensive investigations into optimal prompt designs that significantly impact data quality and generation efficiency.

In response, this study examines key components of prompt design and identifies an effective method for generating high-quality synthetic tabular data, particularly addressing class imbalance. We introduce a novel approach, **EPIC**, which leverages the in-context learning capabilities of LLMs to produce synthetic tabular data with balanced class representation. EPIC incorporates prompt design strategies such as CSV style formatting, balanced class grouping, and a unique variable mapper, which together contribute to generating synthetic data that accurately represents class-specific distributions and feature correlations, as illustrated in Fig. 1.

Extensive evaluations across six real-world datasets demonstrate that EPIC significantly improves the data quality and generation efficiency, achieving state-of-the-art performance. As shown in Fig. 2, baselines exhibit low sensitivity, struggling to accurately generate minority class samples due to inherent class imbalances. In contrast, EPIC achieves high sensitivity and balanced performance for all classes, underscoring its robustness and practicality for real-world applications.

Figure 3: **Overview of our approach.** Our prompt includes repeated data example sets consisting of feature names and class-balanced groups, with the feature name at the end serving as a trigger for the LLM to generate realistic synthetic tabular data. The proposed unique variable remaps categorical values to distinct alphanumeric strings, ensuring clear distinction and variability among variables.

In summary, our key contributions are as follows:

- Our study explores the effectiveness of LLMs in generating realistic synthetic tabular data through in-context learning, providing prompt design guidelines to efficiently generate high-quality data while addressing class imbalance.

- We propose EPIC, a simple yet effective prompting method that uses balanced, grouped data samples with unique variable mapping to generate tabular data that accurately represents both minority and majority classes, preserving feature correlations and overall data distribution.

- The proposed approach is model-agnostic, generally applicable to various LLMs, and easy to implement for any tabular data with minimal preprocessing requirements.

- Extensive experiments on six real-world public tabular datasets and one toy dataset demonstrate the effectiveness of our approach, significantly improving ML classification performance and data generation efficiency.

By addressing class imbalance and enhancing classification outcomes, our work contributes to the advancement of the crucial field of tabular data research, significantly impacting various domains.

## 2 Method

In this study, we investigate various prompt design components to maximize the in-context learning capabilities of LLMs for generating high-quality tabular data. Our objective is to develop an optimized approach that reliably and effectively produces realistic tabular data, accurately representing both minority and majority classes to improve ML classification performance, especially addressing class imbalance. Formally, given a tabular dataset $T$ of dimensions $n \times m$, with $n$ samples and $m$ variables, we aim to generate a synthetic dataset $\hat{T}$ of dimensions $n' \times m$ that accurately reflects key characteristics of $T$, including class-specific attributes, feature correlations and, data distributions.

To achieve this, we explore the following key prompt design elements: data format (Section 2.1), class presentation methods (Section 2.2), variable mapping (Section 2.3), and task specification (Section 2.4). These design choices are evaluated through extensive analyses on the public datasets

from diverse domains, focusing on ML classification performance and generation efficiency. Detailed analysis is provided in Section 3, with Fig. 12 in Appendix C.1 illustrating the ablated versions.

Building on these analyses, we introduce **EPIC**, a structured prompting method designed to guide LLMs in synthetic data generation while effectively addressing class imbalance, as depicted in Fig. 3. EPIC begins with optional variable descriptions, followed by a series of structured data sample sets, each containing balanced samples for each class. Each set consists of feature names and samples organized by target class. To prompt the LLM to generate a corresponding set of synthetic samples, feature names are appended at the end of the prompt to serve as a trigger, leveraging the pattern recognition capabilities of LLMs. The following sections detail each design component of EPIC.

## 2.1 Data format: Sentence vs CSV style

We investigate effective data formatting methods to represent tabular data within prompts, ensuring that LLMs can interpret the demonstrations and generate new data accurately. Tabular data can be formatted as plain text or as comma-separated values (CSV style). Let the feature name of the $k$-th variable be $v_k$, and the $i$-th observation of this variable be $o_{k,i}$. According to previous work [4], this value can be transformed into a sentence-style representation as $[v_k, "is", o_{k,i}, ","]$. In contrast, the CSV style can be expressed as $[o_{k,i}, ","]$, presenting only values, with variable names for all variables specified as $\boldsymbol{v} = [v_1, ",", v_2, ",", \cdots, v_m]$ at the beginning of the data sample presentation. Here, the sentence style redundantly uses variable names and "is" tokens for every value, leading to higher token usage and computational cost than the CSV style. The CSV-style format enables a higher volume of in-context learning examples within the same token constraints. Thus, our method utilizes a CSV-style format to maximize the number of examples, as providing a large number of data samples is crucial for ensuring sufficient representation of the original dataset in in-context learning.

## 2.2 Class presentation

Class presentation methods aim to ensure adequate representation of all target classes within a prompt, particularly addressing underrepresented classes in imbalanced datasets. The construction of data samples within the prompt is crucial, as it can either worsen or mitigate existing imbalances. To address this, we explore three prompt design options and propose a class balancing and grouping approach that enables accurate and representative generation of data samples across all target classes.

**Single-class vs. Multi-class generation**   When generating data under class conditions, a primary consideration is whether to generate data for one class at a time or for all classes simultaneously. Single-class generation allows the model to focus on the unique characteristics of a specific class, while multi-class generation enables direct comparison across classes. Our experiments indicate that generating data for multiple classes simultaneously in a structured manner (discussed in the following sections) yields samples that more accurately capture the distinctive features of each class. This finding suggests that generating data with contextual awareness of other class characteristics enhances the representativeness of synthetic samples.

**Original vs. Balanced class ratio**   When constructing prompts to generate multiple classes simultaneously, simple random sampling from the original dataset $T$ often results in an imbalanced distribution of samples across target classes. This imbalance may cause the LLM to overfit to the majority class, limiting its ability to learn the characteristics of the minority class. To mitigate this, we employ a balanced sampling approach, equalizing sample numbers for all classes rather than strictly following the original class distribution. Our findings reveal that balancing sample sizes substantially improves the quality of generated data for minority classes, indicating that providing balanced data enables LLMs to learn more effectively and replicate the accurate characteristics of all target classes.

**Listing vs. Class grouping**   Within a prompt, data samples can either be presented sequentially, in the order they are sampled, or grouped by class. Grouping emphasizes contrasts between different classes and reinforces similarities within each group, enabling the LLM to generate more distinct and coherent samples for each group. Empirical evidence suggests that grouping yields a better representation of class-specific feature correlations in the generated data. Class grouping also contributes to constraining the LLM to generate data with specific attributes more efficiently. Without grouping, guiding the LLM to produce samples with targeted attributes can be challenging, often requiring numerous iterations to achieve the desired outcome. In contrast, grouping samples by specific features facilitates conditioning the LLM to generate data in groups with particular attributes or conditions. Although

this work primarily focuses on class conditions, this approach can also be applied to feature grouping, enabling controlled generation based on selected criteria.

## 2.3 Variable mapping: Original values vs. Unique variable mapping

Unlike existing methods that often require extensive preprocessing and handling of noisy or missing data [19, 40], our approach minimizes these steps, preserving the integrity of the raw data, including original feature names and values, similar to other LLM-based approaches [4]. However, a critical challenge arises when a dataset contains numerous categorical variables with identical values (e.g., extensive boolean variables). In such cases, data examples in the prompt may become monotonous and repetitive, making it difficult for the model to distinguish between variables, which can significantly degrade the quality of the generated data. Moreover, when faced with such repetitive input, LLMs tend to struggle to generate valid samples, leading to a notable decline in generation efficiency.

To address this, we propose a unique variable mapping method, which remaps the values of each categorical variable to distinct random alphanumeric strings, as illustrated in Fig. 3. Uniform substitution is applied across the dataset, maintaining the integrity of the data structure while introducing necessary variation. For example, consider a categorical variable $v_1$ with values of 't' and 'f'. Our approach remaps these values to distinct three-character strings, such as 'JY0' and 'GGN,' respectively, and applies them consistently across all data points in the dataset. Other categorical variables are similarly remapped to unique values. The transformed dataset is then used as in-context learning demonstrations. Although these new values do not hold inherent semantic meaning, they effectively represent categorical distinctions as symbols, allowing the LLM to identify and utilize patterns within them. This approach ensures that each variable has a unique representation, making the variables clearly distinguishable and leading to more accurate and efficient data generation.

## 2.4 Task specification

A crucial aspect of the prompt design is clearly guiding the LLM to generate synthetic data as intended. To achieve this, we consider two options and ultimately find that prompting the LLM to learn the pattern from a structured, consistent format with optional descriptions for variables is more effective for reliable generation than providing explicit instructions.

**Explicit instruction vs. Completion triggering**     To clarify the task for the LLM, a prompt might include explicit instructions, such as "generate new data samples." However, crafting specific instructions can be challenging, as LLMs are highly sensitive to subtle prompt variations, making it inefficient to determine an optimal phrasing for tabular data generation. A simpler and more effective approach is to provide patterns within the prompt for the LLM to mimic by formatting data samples without explicit instructions, leveraging its advanced pattern recognition capabilities. Formally, let the $i$-th set of data samples across all $c$ classes be structured as $\boldsymbol{S}_i = [\boldsymbol{v}, \boldsymbol{s}_{i,1}, \cdots, \boldsymbol{s}_{i,c}]$, where each $\boldsymbol{s}_{i,k}$ contains $n$ samples for the $k$-th class. Here, $\boldsymbol{v}$ represents feature names and serves as a header to indicate the start of a set. This approach enables the LLM to recognize the pattern in which each set of data samples is structured and consistently begins with $\boldsymbol{v}$. We then input $t$ such sets in a prompt, $[\boldsymbol{S}_1, \boldsymbol{S}_2, \ldots, \boldsymbol{S}_t]$, to establish a recognizable pattern, expecting the LLM to generate the next set, $\boldsymbol{S}_{t+1} = [\boldsymbol{v}, \boldsymbol{s}_{t+1,1}, \cdots, \boldsymbol{s}_{t+1,c}]$. To reinforce this, we place a trigger at the end of the prompt by including the new header $[\boldsymbol{v}]$ for $\boldsymbol{S}_{t+1}$, signaling the LLM to complete the sequence with $[\boldsymbol{s}_{t+1,1}, \cdots, \boldsymbol{s}_{t+1,c}]$. By structuring the input prompt in this way, we guide the LLM to generate a consistent number of samples, $n$, for each target class.

**Providing contextual information**    Optionally, adding contextual information to the prompt can help the LLM understand the characteristics of the original dataset and generate accurate samples. Therefore, when available, we include line-by-line variable descriptions at the beginning of the prompt. Additionally, we treat the target class variable as one of the attributes and position it as the first variable to clarify that each group is organized by the target class.

## 2.5 Overview of tabular data generation process

The synthetic data generation process begins by creating a prompt template and determining whether the unique variable mapping is needed. The prompt template specifies parameters such as the target class, sample size per group, the number of groups, and the number of sets to construct demonstrations.

We then randomly sample data examples from the dataset to populate the template, ensuring that examples do not overlap within each prompt to maintain diversity. Using the provided examples, the LLM generates new data in a CSV format, which is subsequently converted into a structured tabular format. Instances containing categorical variable values not present in the original dataset are discarded to maintain authenticity. This generation process is repeated until the desired sample size is achieved, with new examples sampled with replacement in each iteration. This approach enables the LLM to encounter diverse combinations of real samples and produce rich synthetic samples (see Appendix B.1 for analysis). Although each individual generation reflects the distribution of the provided subset, successive iterations enable the LLM to cover a broader range of the original data cumulatively. This iterative method ensures that the output generated by the LLM comprehensively reflects the characteristics of the original data. Final prompting examples are available in Appendix C.5

## 3 Experiments

This section describes the experimental setup and provides a comprehensive evaluation of our method, assessing its effectiveness across diverse real-world datasets.

**Summary** We conduct three unique experiments to analyze the utility of our method in enhancing ML classification performance: augmenting the original dataset with generated data (Tables 1, 19, Figs. 2, 14), augmenting only the minority class similar to SMOTE (Table 6), and using only generated data (Table 7). Additionally, we evaluate our approach using three LLMs: Mistral, Llama2, and gpt3.5-turbo (Table 2, Figs. 4, 5). We perform ablations on prompt elements to compare classification performance (Tables 3, 5) and conduct unique analyses of token usage and LLM generation efficiency (Tables 4, 8, 9). To examine feature correlations, we separately analyze minority and majority classes, comparing results across all prompt variations (Figs. 4, 5). We further investigate how varying the number of generated samples affects classification performance (Figs. 6, 7). Using a toy dataset, we analyze how different prompts influence the accuracy of generated data distributions (Figs. 1, 9). Lastly, we explore the sampling of input examples and their corresponding outputs in LLMs using the toy dataset, providing insights into the variability and reliability of generated data (Figs. 10, 11).

### 3.1 Experimental setup

**Datasets** We evaluate our method using six real-world public tabular classification datasets from diverse domains: Travel (Marketing), Sick (Healthcare), HELOC (Finance), Income (Social science), Diabetes (Healthcare), and Thyroid (Healthcare). The Thyroid dataset, released after the training cut-off date for GPT-3.5-turbo-0613, provides a more rigorous validation of our approach on completely unseen data. For binary classification datasets, the minority class is designated as one. All duplicate data samples are removed from the datasets. Each dataset is randomly split into 80% training and 20% test sets. We retain the original data, including missing or noisy features. The exception is the Sick dataset, where we follow the source's method. Further details are provided in Appendix C.2.

**Evaluation measure** We evaluate our method based on ML classification performance, generation efficiency, feature correlation, and analysis of generated data distribution. For classification, we report F1 score, sensitivity, specificity, and balanced accuracy (BAL ACC). Feature correlation is measured using Pearson correlation for numerical variables and Cramér's V correlation for categorical variables.

**Baseline models** In our study, we compare our method with various generative models for tabular data, including SMOTE [5], SMOTENC [5], TVAE [35], CopulaGAN [22], CTGAN [35], CTAB-GAN [39], CTAB-GAN+ [40], GReaT [4], and TabDDPM [19]. To compare prompting methods, we also use the prompts from CuratedLLM [27] and LITO [37].

**Experimental details** Our method utilizes the GPT-3.5 models (GPT-3.5-turbo-0613 and GPT-3.5-turbo-16k-0613), Mistral-7b-v0.1 [15], and Llama-2-7b [29]. Unless stated otherwise, we use the GPT-3.5 model for our method. The number of synthetic data samples is based on the size of the original datasets. Across all experiments, unless stated otherwise, we report results from four top-performing ML classifiers: XGBoost [7], CatBoost [24], LightGBM [17], and Gradient boosting classifier [10], known for their strong performance, often surpassing recent deep learning models on tabular datasets. Each classifier is executed in five independent runs, with results averaged over a total of 20 runs to ensure robustness. Further details are available in Appendix C.

Table 1: **Comparison of ML classification performance when synthetic data are added to the original dataset.** Results are averaged across four classifiers, with each model run five times. Complete results, including all baselines and standard deviation values, are provided in Table 19.

| Dataset | Method | #syn | F1 score ↑ | BAL ACC ↑ | Sensitivity ↑ | Specificity ↑ |
|---------|--------|------|------------|-----------|---------------|---------------|
| Travel | Original | - | 58.12 (0.00) | 71.00 (0.00) | 57.00 (0.00) | 85.00 (0.00) |
| | +TVAE [35] | +1K | 59.78 (+1.66) | 72.35 (+1.35) | 62.00 (+5.00) | 82.69 (-2.31) |
| | +CopulaGAN [22] | +1K | 21.76 (-36.36) | 55.52 (-15.48) | 12.80 (-44.20) | 98.23 (+13.23) |
| | +CTAB-GAN+ [40] | +1K | 54.66 (-3.46) | 68.62 (-2.38) | 53.00 (-4.00) | 84.23 (-0.77) |
| | +GReaT [4] | +1K | 60.95 (+2.83) | 72.86 (+1.86) | 58.80 (+1.80) | 86.92 (+1.92) |
| | +TabDDPM [19] | +1K | 53.20 (-4.92) | 67.70 (-3.30) | 50.40 (-6.60) | 85.00 (0.00) |
| | **+Ours** | +1K | **66.65 (+8.53)** | **78.23 (+7.23)** | **78.00 (+21.00)** | 78.46 (-6.54) |
| Sick | Original | - | 87.81 (0.00) | 91.22 (0.00) | 82.83 (0.00) | 99.61 (0.00) |
| | +TVAE [35] | +1K | 87.77 (-0.04) | 91.47 (+0.25) | 83.37 (+0.54) | 99.56 (-0.05) |
| | +CopulaGAN [22] | +1K | 83.60 (-4.21) | 86.61 (-4.61) | 73.37 (-9.46) | 99.86 (+0.25) |
| | +CTAB-GAN+ [40] | +1K | 82.35 (-5.46) | 86.28 (-4.94) | 72.83 (-10.00) | **99.74 (+0.13)** |
| | +GReaT [4] | +1K | 87.23 (-0.58) | 90.83 (-0.39) | 82.07 (-0.76) | 99.60 (-0.01) |
| | +TabDDPM [19] | +1K | 85.17 (-2.64) | 89.30 (-1.92) | 79.02 (-3.81) | 99.57 (-0.04) |
| | **+Ours** | +1K | **88.71 (+0.90)** | **92.93 (+1.71)** | **86.41 (+3.58)** | 99.44 (-0.17) |
| HELOC | Original | - | 71.01 (0.00) | 73.21 (0.00) | 67.89 (0.00) | 78.52 (0.00) |
| | +TVAE [35] | +1K | 71.12 (+0.11) | 73.25 (+0.04) | 68.15 (+0.26) | 78.34 (-0.18) |
| | +CopulaGAN [22] | +1K | 71.23 (+0.22) | 73.32 (+0.11) | 68.37 (+0.48) | 78.26 (-0.26) |
| | +CTAB-GAN+ [40] | +1K | 71.03 (+0.02) | 73.15 (-0.06) | 68.13 (+0.24) | 78.17 (-0.35) |
| | +GReaT [4] | +1K | 70.35 (-0.66) | 72.96 (-0.25) | 66.22 (-1.67) | **79.70 (+1.18)** |
| | +TabDDPM [19] | +1K | 70.65 (-0.36) | 72.89 (-0.32) | 67.51 (-0.38) | 78.26 (-0.26) |
| | **+Ours** | +1K | **71.92 (+0.91)** | **73.66 (+0.45)** | **69.96 (+2.07)** | 77.35 (-1.17) |
| Income | Original | - | 66.90 (0.00) | 76.45 (0.00) | 57.28 (0.00) | **95.61 (0.00)** |
| | +TVAE [35] | +20K | 66.96 (+0.06) | 76.80 (+0.35) | 59.13 (+1.85) | 94.48 (-1.13) |
| | +CopulaGAN [22] | +20K | 66.75 (-0.15) | 76.73 (+0.28) | 59.16 (+1.88) | 94.29 (-1.32) |
| | +CTAB-GAN+ [40] | +20K | 66.49 (-0.41) | 76.42 (-0.03) | 58.14 (+0.86) | 94.70 (-0.91) |
| | +GReaT [4] | +20K | 67.95 (+1.05) | 77.51 (+1.06) | 60.69 (+3.41) | 94.33 (-1.28) |
| | +TabDDPM [19] | +20K | 66.85 (-0.05) | 76.50 (+0.05) | 57.70 (+0.42) | 95.30 (-0.31) |
| | **+Ours** | +20K | **69.16 (+2.26)** | **79.15 (+2.70)** | **66.45 (+9.17)** | 91.85 (-3.76) |
| Diabetes | Original | - | 54.87 (0.00) | 42.07 (0.00) | 60.00 (0.00) | 60.73 (0.00) |
| | +TVAE [35] | +10K | 54.79 (-0.08) | 41.96 (-0.11) | 59.96 (-0.04) | 60.71 (-0.02) |
| | +CopulaGAN [22] | +10K | 54.27 (-0.60) | 41.59 (-0.48) | 59.73 (-0.27) | 59.97 (-0.76) |
| | +CTAB-GAN+ [40] | +10K | 54.24 (-0.63) | 41.52 (-0.55) | 59.63 (-0.37) | 60.01 (-0.72) |
| | +GReaT [4] | +10K | 54.78 (-0.09) | 41.98 (-0.09) | 59.98 (-0.02) | 60.61 (-0.12) |
| | +TabDDPM [19] | +10K | 54.64 (-0.23) | 41.83 (-0.24) | 59.91 (-0.09) | 60.55 (-0.18) |
| | **+Ours** | +10K | **54.94 (+0.07)** | **42.14 (+0.07)** | **60.04 (+0.04)** | **60.82 (+0.09)** |
| Thyroid | Original | - | 94.23 (0.00) | 95.08 (0.00) | 91.14 (0.00) | 99.02 (0.00) |
| | +TVAE [35] | +1K | 90.45 (-3.78) | 92.20 (-2.88) | 86.36 (-4.78) | 98.04 (-0.98) |
| | +CopulaGAN [22] | +1K | 86.73 (-7.50) | 88.71 (-6.37) | 78.41 (-12.73) | 99.02 (0.00) |
| | +CTAB-GAN+ [40] | +1K | 27.46 (-66.77) | 58.07 (-37.01) | 16.14 (-75.00) | **100.0 (+0.98)** |
| | +GReaT [4] | +1K | 91.31 (-2.92) | 92.46 (-2.62) | 85.91 (-5.23) | 99.02 (0.00) |
| | +TabDDPM [19] | +1K | 94.39 (+0.16) | 96.26 (+1.18) | **95.45 (+4.31)** | 97.06 (-1.96) |
| | **+Ours** | +1K | **94.80 (+0.57)** | **96.39 (+1.31)** | 95.23 (+4.09) | 97.55 (-1.47) |

## 3.2 Machine learning classification performance using the synthetic data

We evaluate the quality of the synthetic data samples by assessing the machine learning classification performance when the synthetic data are added to the original dataset. Here, our method utilizes the GPT-3.5-turbo model. As shown in Table 1, our method **achieves state-of-the-art F1 scores and balanced accuracy across all six datasets**, surpassing the baselines that require model training. Notably, our method is **the only one that consistently outperforms the original data in both F1 score and balanced accuracy across all six datasets.** For a machine learning model to perform well in classification, the correlation between input data and labels in the training data must be precise. Thus, the consistent improvement across six datasets demonstrates that the data generated by our method aligns well with the class labels. Other methods often result in worse performance than the

Table 2: **Comparison of classification performance using synthetic data generated by different LLMs with our method.** We report the average performances of five runs of gradient boosting classifier. #syn denotes the number of synthetic samples added to the original dataset.

| Dataset | Method | Model | #syn | F1 score ↑ | BAL ACC ↑ | Sensitivity ↑ | Specificity ↑ |
|---------|--------|-------|------|-----------|-----------|---------------|---------------|
| Travel | Original | - | - | $60.00_{\pm0.00}$ | $72.31_{\pm0.00}$ | $60.00_{\pm0.00}$ | **$84.62_{\pm0.00}$** |
| | +**Ours** | Mistral | +1K | $66.67_{\pm0.00}$ | $78.00_{\pm0.00}$ | $76.00_{\pm0.00}$ | $80.00_{\pm0.00}$ |
| | +**Ours** | Llama2 | +1K | $67.80_{\pm0.00}$ | $79.23_{\pm0.00}$ | **$80.00_{\pm0.00}$** | $78.46_{\pm0.00}$ |
| | +**Ours** | GPT3.5 | +1K | **$70.18_{\pm0.00}$** | **$80.77_{\pm0.00}$** | **$80.00_{\pm0.00}$** | $81.54_{\pm0.00}$ |
| Sick | Original | - | - | $84.09_{\pm0.00}$ | $89.86_{\pm0.00}$ | $80.43_{\pm0.00}$ | $99.28_{\pm0.00}$ |
| | +**Ours** | Mistral | +1K | $88.42_{\pm0.00}$ | **$95.15_{\pm0.00}$** | **$91.30_{\pm0.00}$** | $99.00_{\pm0.00}$ |
| | +**Ours** | Llama2 | +1K | $85.53_{\pm0.95}$ | $93.14_{\pm0.52}$ | $87.39_{\pm0.97}$ | $98.88_{\pm0.06}$ |
| | +**Ours** | GPT3.5 | +1K | **$91.95_{\pm0.00}$** | $93.41_{\pm0.00}$ | $86.96_{\pm0.00}$ | **$99.86_{\pm0.00}$** |
| HELOC | Original | - | - | $71.32_{\pm0.04}$ | $73.39_{\pm0.03}$ | $68.47_{\pm0.06}$ | $78.31_{\pm0.00}$ |
| | +**Ours** | Mistral | +1K | $71.48_{\pm0.00}$ | **$73.74_{\pm0.00}$** | $68.00_{\pm0.00}$ | **$79.47_{\pm0.00}$** |
| | +**Ours** | Llama2 | +1K | $70.77_{\pm0.00}$ | $73.08_{\pm0.00}$ | $67.37_{\pm0.00}$ | $78.79_{\pm0.00}$ |
| | +**Ours** | GPT3.5 | +1K | **$71.93_{\pm0.02}$** | $73.68_{\pm0.02}$ | **$69.90_{\pm0.00}$** | $77.45_{\pm0.04}$ |

original data, indicating that the data generated by baselines disrupts the original data distribution. These findings underscore the robustness and effectiveness of our method, affirming its superiority over baselines, even for challenging datasets.

Moreover, baselines such as GReaT and TabDDPM exhibit significantly lower sensitivity compared to specificity, particularly for Travel and Income. Their high balanced accuracy is due to high specificity, but their severely low sensitivity indicates a failure to generate appropriate minority class data. In contrast, our method **significantly improves sensitivity and achieves a balance between sensitivity and specificity while also attaining the best-balanced accuracy and F1 score compared to the baselines**. For example, in Travel, our method achieves a sensitivity of 78%, which is 19%p higher than the second-best method, GReaT, at 58.8%. In Income, our method shows 66.45% sensitivity, surpassing the second-best score of 60.69%. These results highlight the effectiveness of our method in accurately representing minority classes in tabular datasets.

Overall, our findings demonstrate that our approach is generally applicable across various imbalanced tabular datasets and excels in generating high-quality samples that improve ML classification performance. Further analyses exploring the impact of augmenting the original dataset for the minority class (Table 6) and replacing the original dataset with synthetic data (Table 7) are available in Appendix A.

## 3.3 Open-source LLMs

We also apply our prompting method to open-source LLMs, including Llama2 [29] and Mistral [15]. As shown in Table 2, our method exhibits robust performance when used with open-source LLMs across various datasets. Notably, in the Sick dataset, Mistral demonstrates the highest balanced accuracy, indicating its effectiveness in generating high-quality tabular data. These results validate the broader applicability and general effectiveness of our approach with open-source LLMs.

## 3.4 Exploring optimal prompt design through ablation studies

We investigate key prompt design choices and validate our method through extensive ablation studies across multiple datasets from diverse domains, focusing on ML performance and generation efficiency, as detailed in Tables 3 and 4, respectively. Given a fixed number of input samples, we evaluated (1) the number of input tokens required, (2) the number of valid generated samples, and (3) the generation success rate. Our results indicate that CSV-style prompting generally outperforms sentence-style prompting in ML performance with the same number of input samples. While generating data for one class at a time yields good results for the Sick and Income datasets, it results in a low F1 score of 58.67% on the Travel dataset. However, generating data for multiple classes simultaneously with balanced samples improves this score to 70.37%. Furthermore, applying unique variable mapping consistently enhances F1 scores on the Sick dataset. The combined approach of random mapping and grouping achieves higher F1 score of 91.95% compared to 86.60% with random mapping but without grouping, underscoring the positive impact of this combined approach. These findings indicate

Table 3: **Comparison of F1 scores with ablated methods using the gradient boosting classifier.** Asterisk (*) in Income indicates where only +1K synthetic samples are used due to a low success rate in sample generation. The '-' symbol denotes where our unique variable mapping is not needed.

| Format | Class | Balance | Group | Unique | Sick (+1K) | Travel (+1K) | Income (+20K) |
|---|---|---|---|---|---|---|---|
| Sentence | single | ✗ | ✗ | ✗ | $81.52_{\pm0.65}$ | $43.24_{\pm0.00}$ | $69.62_{\pm0.00}$ |
| CSV-style | single | ✗ | ✗ | ✗ | $86.60_{\pm0.00}$ | $58.67_{\pm0.00}$ | $70.49_{\pm0.00}$ |
| Sentence | multi | ✗ | ✗ | ✗ | $77.95_{\pm0.33}$ | $58.33_{\pm0.00}$ | $67.96_{\pm0.00}$ |
| CSV-style | multi | ✗ | ✗ | ✗ | $87.74_{\pm0.96}$ | $50.98_{\pm0.00}$ | $^{*}68.56_{\pm0.00}$ |
| Sentence | multi | ✓ | ✗ | ✗ | $81.68_{\pm0.49}$ | $55.56_{\pm0.00}$ | $69.26_{\pm0.00}$ |
| CSV-style | multi | ✓ | ✗ | ✗ | $84.68_{\pm0.38}$ | $\mathbf{70.37}_{\pm0.00}$ | $\mathbf{70.64}_{\pm0.00}$ |
| CSV-style | multi | ✓ | ✗ | ✓ | $86.60_{\pm0.00}$ | - | - |
| CSV-style | multi | ✓ | ✓ | ✗ | $81.55_{\pm0.00}$ | $70.18_{\pm0.00}$ | $70.17_{\pm0.00}$ |
| CSV-style | multi | ✓ | ✓ | ✓ | $\mathbf{91.95}_{\pm0.00}$ | - | - |

Table 4: **Comparison of token usage and generation efficiency across ablated methods on the Income dataset.** Results are based on 100 inferences, each with 20 random input samples. `Input tokens` indicates the number of tokens required in the LLM prompt for a fixed number of input samples. `Output samples` shows the average number of synthetic samples generated per iteration. `Success rate` measures the ratio of inferences that generate at least one valid data sample.

| Format | Class | Balance | Group | Unique | #set | Input tokens ↓ | Output samples ↑ | Success rate ↑ |
|---|---|---|---|---|---|---|---|---|
| Sentence | single | ✗ | ✗ | ✗ | - | 1832.9 | 1.26 | 96% |
| CSV-style | single | ✗ | ✗ | ✗ | - | 1024.6 | 1.11 | 16% |
| Sentence | multi | ✗ | ✗ | ✗ | - | 1827.4 | 0.97 | **97%** |
| CSV-style | multi | ✗ | ✗ | ✗ | - | **912.8** | 0.38 | 38% |
| Sentence | multi | ✓ | ✗ | ✗ | - | 1938.7 | 1.73 | **97%** |
| CSV-style | multi | ✓ | ✗ | ✗ | - | 920.3 | 0.88 | 31% |
| CSV-style | multi | ✓ | ✓ | ✗ | 1 | 965.6 | 8.6 | 48% |
| CSV-style | multi | ✓ | ✓ | ✗ | 2 | 1014.6 | **10.52** | 94% |

Table 5: **Comparison of classification performance on the Sick dataset for task specification elements in prompt design.** Results are averaged across XGBoost, CatBoost, LightGBM, and gradient boosting classifier. Methods marked with an asterisk (*) use the prompt designs proposed in the respective papers. #syn denotes the number of synthetic samples added to the original dataset.

| Method | #syn | F1 score ↑ | BAL ACC ↑ | Sensitivity ↑ | Specificity ↑ |
|---|---|---|---|---|---|
| Instruction-CuratedLLM* | +1K | $15.86_{\pm5.13}$ | $56.02_{\pm3.54}$ | $17.17_{\pm11.80}$ | $94.86_{\pm4.93}$ |
| Instruction-LITO* | +1K | $21.32_{\pm1.65}$ | $72.06_{\pm2.49}$ | $84.46_{\pm3.10}$ | $59.66_{\pm3.50}$ |
| Ours w/ class distinction | +1K | $74.06_{\pm2.64}$ | $96.23_{\pm0.98}$ | $\mathbf{96.74}_{\pm1.93}$ | $95.72_{\pm0.60}$ |
| Ours w/o var description | +1K | $76.06_{\pm4.56}$ | $\mathbf{96.29}_{\pm1.40}$ | $96.41_{\pm2.76}$ | $96.18_{\pm1.06}$ |
| **Ours** | +1K | $\mathbf{88.71}_{\pm1.98}$ | $92.93_{\pm0.91}$ | $86.41_{\pm1.85}$ | $\mathbf{99.44}_{\pm0.27}$ |

that prompt design significantly influences the quality of generated data, emphasizing the need for carefully tailored prompts to generate high-quality synthetic tabular data.

For the Travel and Income datasets, performance without grouping is slightly better than with grouping, but the differences are within 0.5%p, making their performance on par. However, these show significant differences in LLM generation efficiency. Without grouping, an average of 0.88 samples are generated per iteration, whereas grouping increases this to an average of 8.6 samples per iteration. Repeating the set twice within the prompt further improves generation efficiency, boosting the average output to over 10 samples and the success rate to 94%, by creating input patterns that LLMs can mimic to generate data. Unique variable mapping also significantly enhances LLM generation efficiency and success rates, as detailed in Table 8 of Appendix A.3.

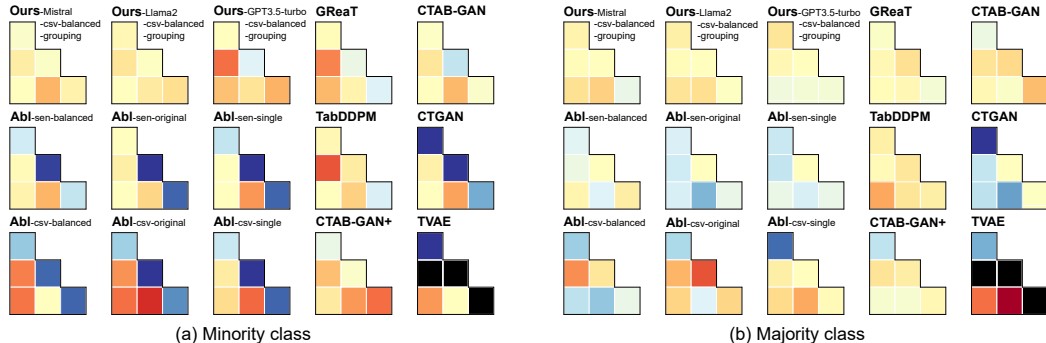

(a) Minority class                 (b) Majority class

Figure 4: **Difference between Cramér's V correlation matrices of real and synthetic datasets for categorical variables in the Travel dataset.** More intense colors indicate larger differences, with positive differences shown in red and negative differences shown in blue. Black indicates where the correlation is not measured since only one unique value is generated for those variables.

Furthermore, we conduct an ablation study on the task specification elements, as shown in Table 5. Our approach of triggering completion significantly enhances overall ML performance compared to the instruction-based prompting methods. Interestingly, adding class distinctions within the prompt achieves the best sensitivity, indicating that the distinction facilitates better generation for minority classes. Providing variable descriptions improves balanced accuracy but reduces the F1 score, suggesting that this element may be optional rather than a definitive improvement. A comparison of generation efficiency using these methods is provided in Table 9 of Appendix A.3.

### 3.5 Analysis of feature correlation

We compare the feature correlations of the categorical variables between original and synthetic data, as illustrated in Fig. 4. Our synthetic data generated with Mistral and Llama2 exhibit the closest feature correlation with the original data for both minority and majority classes, outperforming the baselines and ablated versions. Additionally, our grouping method significantly contributes to preserving feature correlation for the minority class. These findings indicate that our method generates data that more accurately matches each class, particularly for underrepresented classes, compared to other methods. Additional results on the Sick dataset are available in Fig. 5 of Appendix A.3.

## 4 Limitations and future work

When the training dataset is large and cannot be fully included in the LLM prompt due to token size limitations, only a subset of the data can be used as examples for generating samples. If these prompt samples do not fully represent the original data distribution, the generated data may be incomplete and of low quality, as LLMs are limited to producing data based only on the patterns present in the input. To overcome this, our method employs multiple rounds of random sampling with replacement to create a synthetic dataset that more comprehensively represents the original distribution, resulting in improved machine learning classification performance (a comprehensive analysis is provided in Appendix B.1). However, this approach still carries the risk that samples may not fully capture the original data distribution. Future research will focus on developing methods to identify key examples that more accurately represent the entire dataset.

## 5 Conclusion

This study demonstrates the effectiveness of using LLMs for synthetic tabular data generation. We introduce EPIC, a simple yet effective solution for generating realistic, high-quality data without additional training, specifically designed to address class imbalance. Our method demonstrates significant improvements over state-of-the-art generation models across six real-world public datasets, generating data with highly accurate feature correlations and significantly improving ML classification accuracy for minority classes. These results underscore the substantial impact of our approach in real-world applications, making a significant contribution to the field of tabular data research.

## Acknowledgements

This work was supported by Institute for Information & communications Technology Promotion(IITP) grant funded by the Korea government(MSIT) (No.RS-2019-II190075 Artificial Intelligence Graduate School Program(KAIST)), Electronics and Telecommunications Research Institute(ETRI) grant funded by the Korean government (24ZB1220, Fundamental Technology Research for Human-Centric Autonomous Intelligent Systems), and the National Supercomputing Center with supercomputing resources including technical support (KSC-2024-CRE-0028).

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

# Appendix

This supplementary material enhances the main manuscript by providing detailed experimental results and additional visualizations.

- **Appendix A** provides additional experimental results on the machine learning classification for imbalanced data, ablation study, and feature correlation analysis.
- **Appendix B** studies the impact of sample size on the machine learning performance and comprehensively analyzes the experiments conducted on the toy dataset.
- **Appendix C** provides comprehensive information on the datasets, baselines, implementation details, and prompt design and examples.
- **Appendix D** provides related work.
- **Appendix E** discusses the broader impacts of our research.
- **Appendix F** presents confusion matrix results and the complete results of Table 1.

# A    Additional experimental results

Our experiments with synthetic tabular data for imbalanced classes follow three main approaches: (1) Adding synthetic data to both minority and majority classes of the existing dataset and evaluating machine learning classification performance. The results of this approach are presented in Table 1 of the main manuscript. (2) Adding synthetic data only to the minority class, with detailed results provided in Appendix A.1. (3) Evaluating classification performance using only synthetic data without the original dataset, as discussed in Appendix A.2. Moreover, Appendix A.3 presents further ablation study results, and Appendix A.4 provides a feature correlation analysis for the sick dataset.

## A.1    Comparative analysis of augmenting the original dataset for the minority class

One of the major advantages of our method is its ability to generate high-quality synthetic data for minority classes, even with limited samples. Similar to oversampling techniques like SMOTE, we add synthetic dataset only to the minority class and evaluate machine learning classification performance. To balance the ratio of majority to minority classes, we add only minority class samples to the original data and then measure the classification performance of the machine learning model. For this task, we additionally investigate the effectiveness of our method compared to SMOTE and SMOTENC [5]. As shown in Table 6, adding minority class samples generated from our method leads to higher

Table 6: **Comparison of binary classification performance when augmenting the minority class with synthetic data to balance class sizes.** Average performance of the gradient boosting classifier is reported over five runs. `#syn` denotes the number of synthetic samples added to the original dataset.

| Dataset | Method | #syn | F1 score ↑ | BAL ACC ↑ | Sensitivity ↑ | Specificity ↑ |
|---|---|---|---|---|---|---|
| Travel | Original | - | 60.00±0.00 | 72.31±0.00 | 60.00±0.00 | **84.62**±0.00 |
| | +SMOTE | +163 | 63.45±1.05 | 75.08±0.69 | 68.00±0.00 | 82.15±1.38 |
| | +SMOTENC | +163 | 62.61±2.54 | 74.74±2.00 | 70.40±3.58 | 79.08±0.84 |
| | +TVAE | +163 | 54.55±0.00 | 68.46±0.00 | 60.00±0.00 | 76.92±0.00 |
| | +CopulaGAN | +163 | 52.46±0.00 | 66.62±0.00 | 64.00±0.00 | 69.23±0.00 |
| | +CTGAN | +163 | 57.63±0.00 | 70.92±0.00 | 68.00±0.00 | 73.85±0.00 |
| | +GReaT | +163 | 62.07±0.00 | 74.46±0.00 | 72.00±0.00 | 76.92±0.00 |
| | **+Ours** | +163 | **72.13**±0.00 | **83.23**±0.00 | **88.00**±0.00 | 78.46±0.00 |
| Income | Original | - | 69.05±0.00 | 78.02±0.00 | 61.03±0.00 | **95.00**±0.00 |
| | +SMOTE | +13,487 | 69.62±0.25 | 79.84±0.23 | 68.86±0.57 | 90.81±0.14 |
| | +SMOTENC | +13,487 | 69.79±0.11 | 80.51±0.06 | 71.80±0.23 | 89.22±0.19 |
| | +TVAE | +13,487 | 68.26±0.00 | 79.12±0.00 | 68.43±0.00 | 89.82±0.00 |
| | +CopulaGAN | +13,487 | 67.76±0.02 | 81.12±0.01 | 80.15±0.03 | 82.09±0.00 |
| | +CTGAN | +13,487 | 68.51±0.00 | 78.82±0.00 | 66.26±0.00 | 91.38±0.00 |
| | +GReaT | +13,487 | 70.04±0.00 | 83.36±0.00 | **85.08**±0.00 | 81.64±0.00 |
| | **+Ours** | +13,487 | **71.09**±0.02 | **83.60**±0.01 | 83.53±0.03 | 83.66±0.00 |

Table 7: **Comparison of classification performance using only synthetic data for training.** Average performance of the gradient boosting classifier is reported over five runs.

| Dataset | Method | #syn | F1 score ↑ | BAL ACC ↑ | Sensitivity ↑ | Specificity ↑ |
|---|---|---|---|---|---|---|
| Travel | Original | - | $60.00_{\pm0.00}$ | $72.31_{\pm0.00}$ | $60.00_{\pm0.00}$ | $84.62_{\pm0.00}$ |
| | TVAE | 1K | $46.51_{\pm0.00}$ | $63.85_{\pm0.00}$ | $40.00_{\pm0.00}$ | $87.69_{\pm0.00}$ |
| | CopulaGAN | 1K | $0.00_{\pm0.00}$ | $49.23_{\pm0.00}$ | $0.00_{\pm0.00}$ | $\mathbf{98.46}_{\pm0.00}$ |
| | CTGAN | 1K | $12.50_{\pm0.00}$ | $50.15_{\pm0.00}$ | $8.00_{\pm0.00}$ | $92.31_{\pm0.00}$ |
| | GReaT | 1K | $62.50_{\pm0.00}$ | $73.85_{\pm0.00}$ | $60.00_{\pm0.00}$ | $87.69_{\pm0.00}$ |
| | **Ours** | 1K | $\mathbf{70.00}_{\pm0.00}$ | $\mathbf{81.23}_{\pm0.00}$ | $\mathbf{84.00}_{\pm0.00}$ | $78.46_{\pm0.00}$ |
| HELOC | Original | - | $71.32_{\pm0.04}$ | $\mathbf{73.39}_{\pm0.03}$ | $68.47_{\pm0.06}$ | $78.31_{\pm0.00}$ |
| | TVAE | 1K | $68.91_{\pm0.10}$ | $71.50_{\pm0.08}$ | $65.28_{\pm0.12}$ | $77.72_{\pm0.07}$ |
| | CopulaGAN | 1K | $69.50_{\pm0.00}$ | $70.82_{\pm0.00}$ | $69.17_{\pm0.00}$ | $72.47_{\pm0.00}$ |
| | CTGAN | 1K | $68.71_{\pm0.02}$ | $71.98_{\pm0.03}$ | $63.36_{\pm0.00}$ | $80.60_{\pm0.05}$ |
| | GReaT | 1K | $64.14_{\pm0.10}$ | $69.54_{\pm0.07}$ | $55.76_{\pm0.15}$ | $\mathbf{83.33}_{\pm0.11}$ |
| | **Ours** | 1K | $\mathbf{71.48}_{\pm0.06}$ | $70.17_{\pm0.05}$ | $\mathbf{78.90}_{\pm0.16}$ | $61.44_{\pm0.16}$ |

F1 scores and balanced accuracy on both Travel and Income datasets, compared to when samples generated by other baselines are added. In the Travel dataset, where the number of minority class samples is substantially low, baselines, especially CopulaGAN and CTGAN, demonstrate markedly low sensitivity, indicating their failure to accurately learn the minority class data distribution. In contrast, our method demonstrates its efficacy in producing high-quality data for minority classes on the Travel dataset. Compared to the Original, it significantly improves the F1 score by 12.13%p, underscoring its superior performance of our approach.

## A.2 Comparative analysis of replacing the original dataset with synthetic data

We compare the classification performance using only synthetic data for training, as shown in Table 7. For both the Travel and HELOC datasets, the classification model trained on synthetic datasets generated by our model outperforms the one trained on original data in terms of F1 score and sensitivity. This improvement in performance suggests that the data generated by our method accurately represents the distribution of the original data. Furthermore, it highlights the effectiveness of our class-balanced generation approach in improving the performance of classification models.

## A.3 Additional ablation study results

This section presents additional ablation study results on the Sick datasets. We evaluate the token efficiency and generation stability of ablated versions based on 100 inferences, each with 20 random input samples. As shown in Table 8, for the Sick dataset, due to its repetitive, monotonous values, the number of generated samples is low, even with grouping. Using a unique variable mapper in such cases significantly increases the number of samples generated per attempt from 6.93 to 17.68. This improvement demonstrates that our approach effectively controls the quantity of generated data samples, ensuring the stable production of synthetic tabular data. Moreover, the ability to generate multiple data samples in a single inference can significantly lower generation time and costs.

We also compare the efficiency and stability of generation when ablating task specification elements in our prompt design, as shown in Table 9. The prompt design from Curated LLM [27], despite requiring the highest number of input tokens, achieves high performance in terms of output samples and success rate. However, the generated data shows relatively low quality, resulting in a low classification performance with an F1 score and balanced accuracy, as discussed in Section 3.4. In contrast, the prompt design of LITO [37] is the most efficient in terms of input tokens but produces the fewest output samples and has the lowest success rate, indicating instability in synthetic data generation.

Our proposed method balances efficiency and performance, using approximately half the input tokens compared to CuratedLLM's prompt while generating a high number of output samples and maintaining a high success rate. Additionally, the synthetic data generated by our method achieves the highest classification performance in terms of F1 score, as shown in Section 3.4. When class distinctions are provided, the average number of output samples decreased from 17.68 to 9.70. This result indicates that it is more robust to distinguish groups with simple numbering, such as A and

Table 8: **Comparison of token usage and generation efficiency across ablated methods on the Sick dataset.** Results are based on 100 inferences, each with 20 random input samples. `Input tokens` indicates the number of tokens required in the LLM prompt for a fixed number of input samples. `Output samples` shows the average number of synthetic samples generated per iteration. `Success rate` measures the ratio of inferences that generate at least one valid data sample.

| Format | Class | Balance | Group | Unique | #set | Input tokens ↓ | Output samples ↑ | Success rate ↑ |
|---|---|---|---|---|---|---|---|---|
| Sentence | single | ✗ | ✗ | ✗ | - | 3867.2 | 0.52 | 52% |
| CSV-style | single | ✗ | ✗ | ✗ | - | 1223.6 | 2.42 | 38% |
| Sentence | multi | ✗ | ✗ | ✗ | - | 3879.9 | 0.52 | 52% |
| CSV-style | multi | ✗ | ✗ | ✗ | - | 1226.3 | 3.05 | 48% |
| Sentence | multi | ✓ | ✗ | ✗ | - | 3870.0 | 0.63 | 63% |
| CSV-style | multi | ✓ | ✗ | ✗ | - | **1117.3** | 0.70 | 16% |
| CSV-style | multi | ✓ | ✗ | ✓ | - | 1985.4 | 3.00 | 15% |
| CSV-style | multi | ✓ | ✓ | ✗ | 1 | 1331.3 | 4.73 | 64% |
| CSV-style | multi | ✓ | ✓ | ✗ | 2 | 1439.9 | 6.93 | **99%** |
| CSV-style | multi | ✓ | ✓ | ✓ | 2 | 2060.8 | **17.68** | 95% |

Table 9: **Comparison of token usage and generation efficiency for task specification ablation on the Sick dataset.** Results are based on 100 inferences, each with 20 random input samples.

| Method | Input tokens ↓ | Output samples ↑ | Success rate ↑ |
|---|---|---|---|
| Instruction-CuratedLLM* | 4677.4 | **21.67** | **95%** |
| Instruction-LITO* | **1277.4** | 4.06 | 57% |
| Ours w/ class distinction | 2222.7 | 9.7 | 91% |
| Ours w/o var description | 2066.9 | 15.96 | 88% |
| **Ours** | 2060.8 | 17.68 | **95%** |

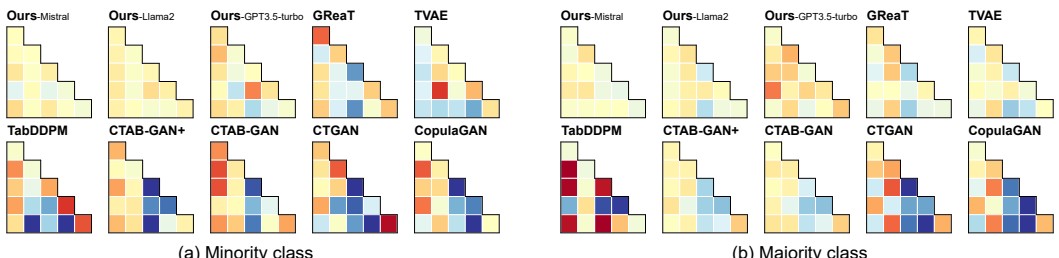

(a) Minority class      (b) Majority class

Figure 5: **Difference between Pearson correlation matrices of real and synthetic data for numerical variables in the Sick dataset.** More intense colors indicate larger differences, with positive differences shown in red and negative differences shown in blue. Black indicates where the correlation is not measured since only one unique value is generated for those variables.

B, rather than explicitly specifying class information in each group. Including variable descriptions slightly reduces the number of output samples and success rate compared to not including them, but the difference is not significant. This suggests that the grouping strategy plays a more crucial role in robust data generation than the presence of variable descriptions.

## A.4 Analysis of feature correlation between numerical variables

In the main manuscript, we visualize the feature correlation between categorical variables. Here, we analyze the Pearson correlation between numerical variables in the Sick dataset and compare the differences in feature correlation values to the original data, as shown in Fig. 5. As a result, our methodology consistently shows feature correlations similar to the original data for both minority and majority classes. In contrast, other baselines either show significant deviations in feature correlation for all classes or perform well for majority classes but not for minority ones. These results demonstrate

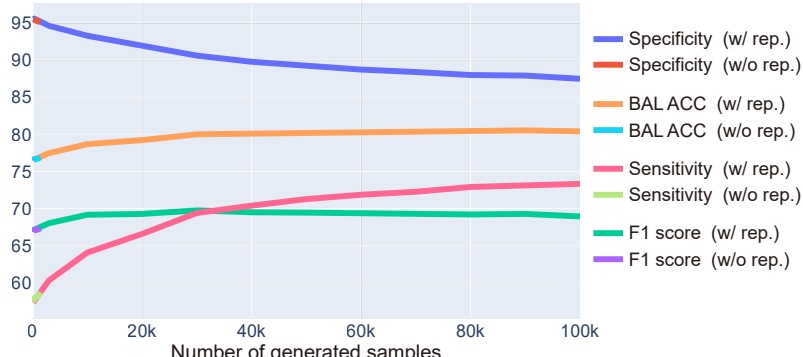

Figure 6: **Classification performance on the Income dataset when synthetic data generated by our proposed method are added to the original dataset.** We experiment with varying synthetic sample sizes, comparing data sampling methods: with replacement (w/ rep.) and without replacement (w/o rep.). EPIC samples with replacement. GPT-3.5-turbo is used for the experiment.

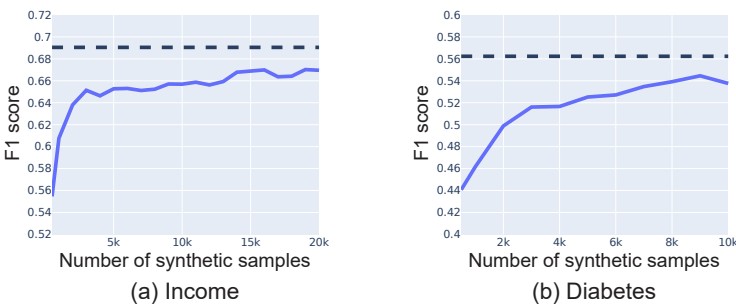

Figure 7: **Classification performance on the Income and Diabetes datasets using only synthetic data generated by our proposed method for training.** Black dashed lines denote the F1 score using only the original data. GPT-3.5-turbo is used for our method.

that our methodology shows superior performance compared to the baselines in generating high-quality data that accurately represents both minority and majority classes within the original dataset.

## B   Additional analysis of EPIC

This section provides an in-depth analysis of the proposed method, EPIC. Appendix B.1 analyzes the effect of sample size on classification performance. Appendix B.2 discusses the significance of the observed performance improvements. Appendix B.3 provides complete qualitative results on the toy dataset. Finally, Appendix B.4 explores tabular data classification with in-context learning methods.

### B.1   Impacts of sample size on the classification performance

**Adding synthetic data to the original dataset**    Data samples in prompts can be selected either with or without replacement. Our method uses sampling with replacement to generate a large and diverse dataset, as sampling without replacement limits the amount of synthetic data to the number of actual data points. While we ensure that there are no overlapping examples within each prompt to maintain diversity, each prompt is constructed using sampling with replacement.

We conduct experiments to analyze (1) how much datasets can be enlarged and (2) how this impacts classification performance by comparing sampling with and without replacement. As shown in Fig. 6, performance improves steadily in both scenarios as the volume of generated data increases. However, the improvement is constrained when sampling without replacement due to the limited number of possible samples. When sampling with replacement, as the dataset size expands, there is a noticeable improvement in the balance between sensitivity and specificity, which contributes to enhanced overall performance, including gains in balanced accuracy and F1 score. Generating up to 40K synthetic

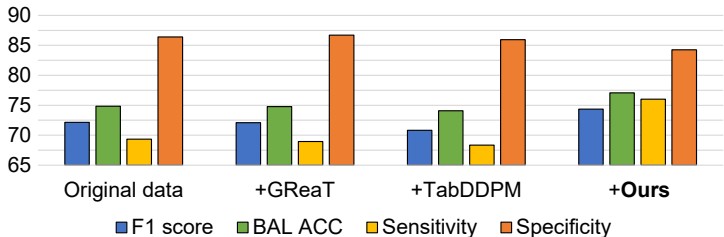

Figure 8: **Average classification results across six datasets compared with the closest baselines.**

Table 10: **Performance improvement after adding synthetic data to the original dataset.** Results are reorganized from Table 1 of our main manuscript.

| Method | Dataset | Improvement from the original data (%p) | | | |
|---|---|---|---|---|---|
| | | F1 score ↑ | BAL ACC ↑ | Sensitivity ↑ | Specificity ↑ |
| GReaT | Travel | 2.83 | 1.86 | 1.8 | 1.92 |
| | Sick | -0.58 | -0.39 | -0.76 | -0.01 |
| | HELOC | -0.66 | -0.25 | -1.67 | 1.18 |
| | Income | 1.05 | 1.06 | 3.41 | -0.31 |
| | Diabetes | -0.09 | -0.09 | -0.02 | -0.12 |
| | Thyroid | -2.92 | -2.62 | -5.23 | 0.00 |
| TabDDPM | Travel | -4.92 | -3.3 | -6.6 | 0.00 |
| | Sick | -2.64 | -1.92 | -3.81 | -0.04 |
| | HELOC | -0.36 | -0.32 | -0.38 | -0.26 |
| | Income | -0.05 | 0.05 | 0.42 | -0.31 |
| | Diabetes | -0.23 | -0.24 | -0.09 | -0.18 |
| | Thyroid | 0.16 | 1.18 | 4.31 | -1.96 |
| **Ours** | Travel | 8.53 | 7.23 | 21 | -6.54 |
| | Sick | 0.9 | 1.71 | 3.58 | -0.17 |
| | HELOC | 0.91 | 0.45 | 2.07 | -1.17 |
| | Income | 2.26 | 2.7 | 9.17 | -3.76 |
| | Diabetes | 0.07 | 0.07 | 0.04 | 0.09 |
| | Thyroid | 0.57 | 1.31 | 4.09 | -1.47 |

data points resulted in even better performance than the 20K synthetic data points reported in Table 1 of our main manuscript. We also observed that as the volume of generated data continues to increase, the gains in balanced accuracy and F1 score eventually plateau, indicating diminishing returns and suggesting that further data generation beyond a certain point offers limited additional benefit.

**Using only synthetic data**  We investigate how varying the number of synthetic samples affects classification performance. This analysis is conducted on the Income and Diabetes datasets, and the results are shown in Fig. 7. For the Income dataset, sample sizes range from 500 to 10,000, while for Diabetes, the range was 500 to 20,000. For both datasets, a consistent increase in the F1 score is observed as the number of samples grows. This result demonstrates that our method can adequately represent the original data distribution by generating a sufficient number of synthetic data samples.

## B.2  Significance of performance improvements

We analyze the significance of performance improvements achieved by EPIC. As shown in Fig. 6, while the gains in F1 score and balanced accuracy between 0 and 40K samples may seem modest, the sensitivity actually increases significantly from around 50% to 70%. This improvement balances performance across all classes, leading to a more meaningful outcome and greatly enhancing model usability. Baselines often learn biases in the training data, resulting in abnormally high specificity at the expense of sensitivity. This imbalance can inflate balanced accuracy or F1 scores, but such performance is ineffective for real-world tasks. In contrast, our method significantly enhances sensitivity with only a slight reduction in specificity. **As a result, our method achieves the most balanced performance across all four metrics**, as shown in Fig. 8.

Our method is the only approach among the seven baselines that consistently improves both F1 score and balanced accuracy compared to using only the original data (Table 19). Extensive experiments across six real-world datasets from diverse domains (finance, healthcare, marketing, and social

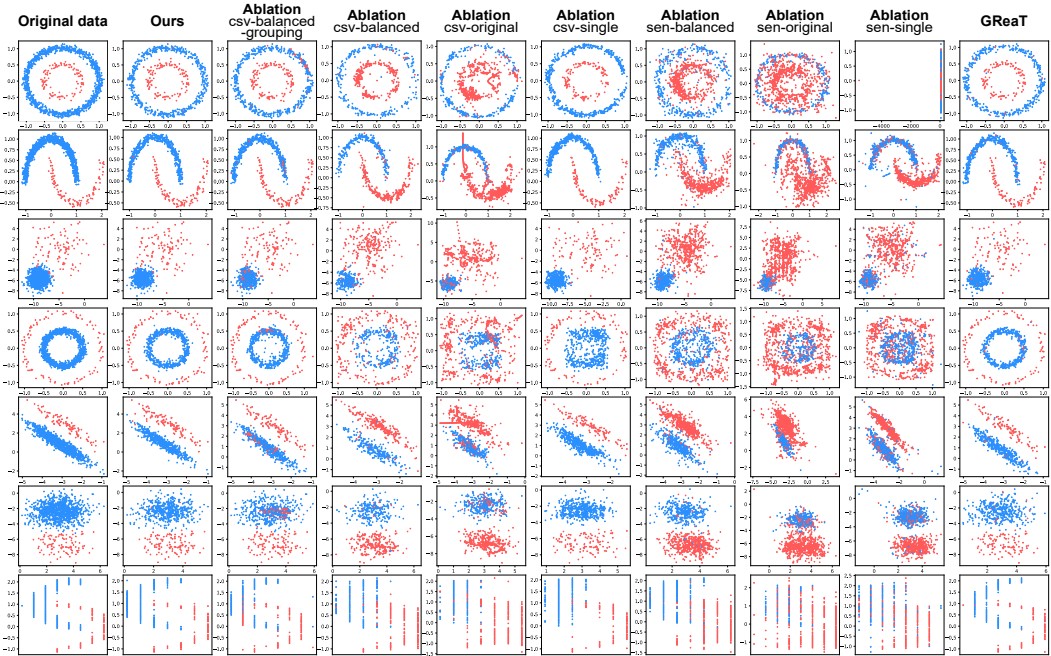

Figure 9: **Complete generation results on an imbalanced toy dataset with majority and minority classes**. Our approach, leveraging in-context learning with LLMs, achieves (1) distinct class boundaries, (2) accurate feature correlations, (3) well-matched value ranges, (4) robust numerical-categorical relationships (last row), and (5) comprehensive data distribution coverage, with improvements over its ablated versions and the fine-tuned GReaT model [4].

science) using four classifiers, each tested five times, demonstrate the state-of-the-art performance of our method, with significant gains in the highly imbalanced Travel, Sick, Income, and Thyroid datasets (Fig. 13). **While GReaT and TabDDPM are the closest baselines, they exhibit inconsistent performance**. As highlighted in Table 10, they reduce original data performance in more than half of the cases (red). For example, both underperform on HELOC and Diabetes.

In stark contrast, **our method consistently outperforms the original data across all datasets** (blue). In the challenging Diabetes dataset, only our method improves over the original data. While specificity decreases in all methods, our method still achieves a superior balance of metrics, leading to greater practicality. These results underscore that our method offers the greatest practical utility and a meaningful performance advantage among the models tested.

## B.3 Complete results on toy dataset

In Fig. 9, we illustrate the complete qualitative results on the toy dataset. We assume a challenging multivariate data scenario in an imbalanced binary classification situation, where variables with and without correlations are mixed. Using the scikit-learn library [23], we created a total of 12 variables. These variables have strong correlations in pairs. There are 11 numerical variables and one categorical variable, and the categorical variable is shown on the x-axis in the last row of Fig. 9. In the figure, the majority class is shown in blue, and the minority class is shown in red.

The generated results are evaluated from four aspects: (1) assessing whether generated data samples correctly belong to their respective classes, ensuring clear class boundaries; (2) examining whether the shapes formed by variable correlations, such as circles or moon patterns, are accurately represented; (3) evaluating whether the generated values fall within an acceptable range for each variable; and (4) analyzing whether relationships between categorical and numerical variables are accurately captured.

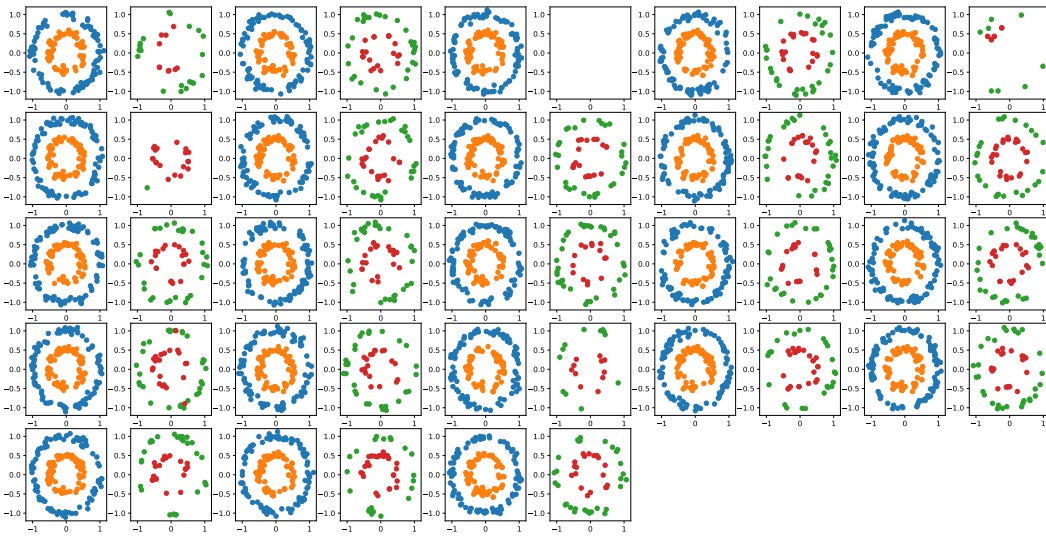

Figure 10: **Illustration of the input samples (majority and minority) and the corresponding generated samples (majority and minority) for each inference of EPIC on the imbalanced toy dataset.** In this example, with 180 input samples per inference, EPIC requires 23 inferences to generate 1,000 samples. Empty boxes represent cases where the model fails to generate valid samples.

The baseline method, GReaT [4], has successfully learned from the training data to generate class-separated samples. However, upon closer inspection, there are samples belonging to different classes. Ours, despite generating samples without training, shows similar or better quality.

Overall, CSV-style generation produces better class separation boundaries compared to sentence-style generation. Regarding class presentation, giving a single class can sometimes result in good generation outcomes, but it fails to accurately capture the correlation with the majority class, resulting in shapes like squares instead of circles, and it also fails to accurately generate the categorical variable. On the other hand, when generating multi-class samples, if the sample numbers are not balanced, the generation results for the minority class can be compromised. Even when balanced, providing grouping leads to clearer class distinctions. Additionally, adding unique variable mapping significantly improves the performance, resulting in a well-distinguished categorical variable.

In addition, we visualize the inputs and outputs of both our method and the ablated model without grouping, as shown in Fig. 10 and Fig. 11. For each inference, 180 class-balanced samples are input into the LLM prompt. To generate a total of 1,000 samples, our method required 23 inferences, whereas the ablated model needed 37 inferences, demonstrating the stability of our method in generating synthetic data samples. In terms of data generation quality, our method successfully generates both majority and minority classes in most inferences, with the distribution of the generated samples closely mirroring that of the actual input data. In contrast, the ablated model exhibits a higher number of inferences where the generated data deviates from the input data distribution.

## B.4 Comparison of advanced tabular classification models

The ability of LLMs to generate high-quality synthetic data for imbalanced classes in our approach highlights their potential to directly address classification tasks, offering a promising avenue for future research. However, the challenge lies in **designing effective prompts to fully leverage this potential**. The performance of LLMs is highly dependent on prompt design, and numerous studies have shown that small variations in prompt crafting can lead to significant differences in outcomes [18, 25]. Our work shares this motivation, focusing on designing prompts that enable LLMs to effectively generate high-quality tabular data, particularly to address class imbalance. While exploring LLMs for direct classification could enhance performance, this direction is beyond the scope of our current study.

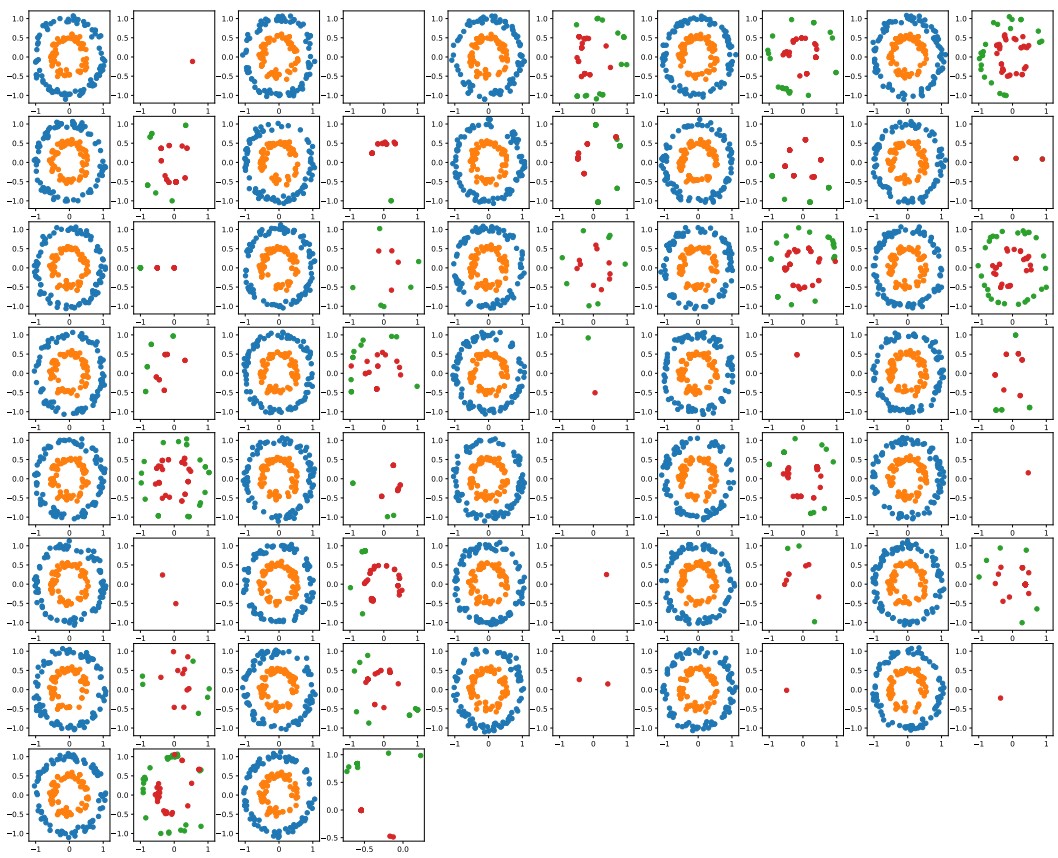

Figure 11: **Illustration of the input samples (majority and minority) and the corresponding generated samples (majority and minority) for each inference of the ablated version of our method on the imbalanced toy dataset.** In this example, with 180 input samples per inference, the ablated method without grouping requires 37 inferences to generate 1,000 samples. Empty boxes represent cases where the model fails to generate valid samples.

We conduct experiments using recent in-context learning-based tabular classification methods, such as TabPFN [14], which employs a pretrained transformer, and T-Table [33], which leverages LLMs. We also test TabR [12], an advanced deep-learning tabular classification model, using its official code. As detailed in Table 11, we evaluate classification performance on the Travel dataset using the original and synthetic data. For T-Table, we use the GPT-3.5-turbo model, incorporating all original data but limiting synthetic samples to 200 per input prompt due to token constraints. To further enhance performance, we employ voting across five inferences [31].

Table 11: **Comparison of F1 scores using robust tabular classification models on the Travel dataset.** GB refers to gradient boosting classifier.

| Model | Original | +Ours | +TabDDPM | +GReaT |
|---|---|---|---|---|
| XGBoost | 55.32 | **67.74** | 51.06 | 64.00 |
| LightGBM | 60.00 | **64.29** | 55.32 | 62.50 |
| CatBoost | 57.14 | **64.41** | 48.09 | 57.73 |
| GB | 60.00 | **70.18** | 58.33 | 59.57 |
| TabPFN | 56.00 | **59.26** | 55.32 | 53.06 |
| T-Table | 21.62 | **30.43** | 10.53 | 17.65 |
| TabR | 46.41 | **60.78** | 44.88 | 32.41 |

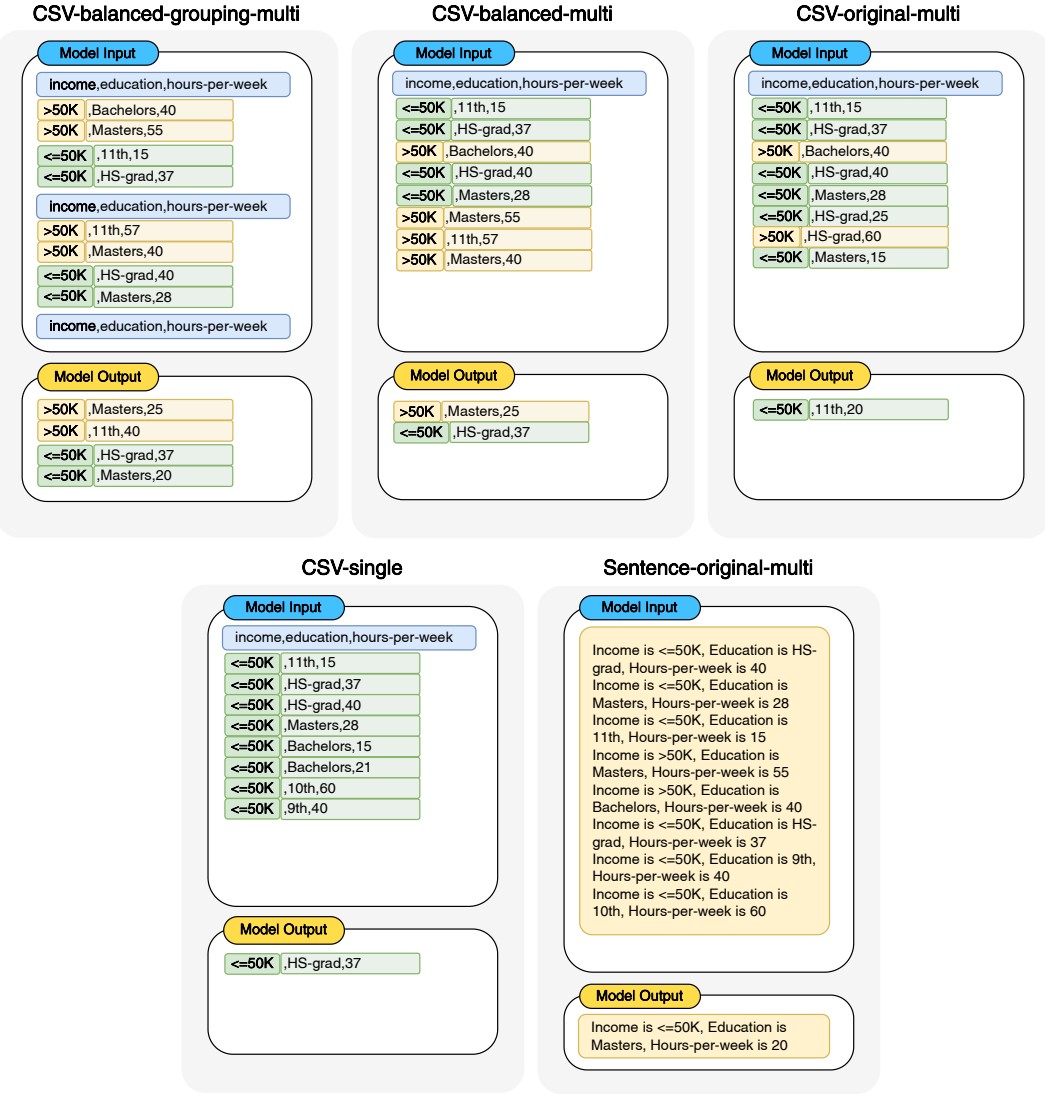

Figure 12: **Illustration of ablated versions of our proposed method.**

The results indicate that, while the newly introduced classifiers do not outperform traditional ones, adding synthetic data generated by our method to the original data consistently leads to significant performance improvements across all classifiers. In contrast, using TabDDPM and GReaT often results in decreased performance. Our method uniquely and consistently enhances the performance of different classifiers, demonstrating superior label-matching quality. These findings underscore **the value of high-quality synthetic data in enhancing classifier performance across diverse models**, highlighting the importance of data generation research as a distinct area from classifier development.

Tabular data generation is a critical area of research with significant implications [4, 27, 37]. This task serves two primary purposes: (1) enhancing classification performance in a model-agnostic manner through data augmentation, similar to SMOTE, as shown in Tables 1, 2, and 6, and (2) generating synthetic data to replace original data in security-critical or privacy-sensitive contexts, as demonstrated in Table 7. In fields such as healthcare, where collecting new samples or achieving balanced class labels is challenging and data may be noisy or incomplete, generating high-quality synthetic data is crucial.

In conclusion, even as more advanced classification models for tabular data are developed in the future, **our proposed method could continue to play a crucial role in enhancing performance**

Table 12: **Dataset details used in this study.**

| Dataset | #Class | #Categorical features | #Numerical features | #Samples | Domain |
|---------|--------|----------------------|---------------------|----------|--------|
| Travel | 2 | 4 | 2 | 447 | Marketing |
| Sick | 2 | 21 | 6 | 3,711 | Medical |
| HELOC | 2 | - | 23 | 9,872 | Financial |
| Income | 2 | 8 | 6 | 32,561 | Social |
| Diabetes | 3 | 36 | 11 | 101,766 | Medical |
| Thyroid | 2 | 16 | 1 | 364 | Medical |

**by generating high-quality synthetic data**, thereby potentially making a significant impact on the tabular data classification community and related tasks.

## C  Additional experimental details

This section provides the illustration of ablated versions (Appendix C.1), comprehensive information on the datasets (Appendix C.2) and comparison baselines (Appendix C.3), implementation details (Appendix C.4), and final prompt design and examples (Appendix C.5).

### C.1  Ablation prompt examples

This section visualizes five ablated versions of our proposed prompt design, as shown in Fig. 12. The ablations involve removing or altering four key components: data format, balanced class ratio, multi-class generation, and class grouping.

### C.2  Dataset

A detailed description of the datasets is shown in Table 12. The variable descriptions are directly sourced from dataset platforms like Kaggle or the UCI repository. In our approach, we designate the class with fewer instances as the positive class for binary classification tasks. We also provide the class distribution information for the datasets used in this work, as shown in Fig. 13. Among the six datasets, the Travel, Income, Sick, and Thyroid datasets exhibit substantial class imbalances.

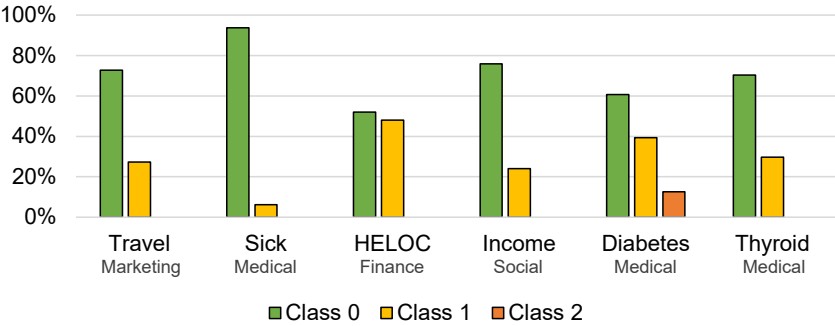

Figure 13: **Class distribution for the real-world public tabular datasets used in this study.**

### C.3  Baseline reproducibility

In our research, we utilize implementations from the Synthetic Data Vault [22] for TVAE [35], CTGAN [35], and CopulaGAN, while for the GReaT [4] model, we employ code from its official repository. In our experiment, the GReaT model fails to accurately generate column names 'max_glu_serum' and 'A1Cresult' in the Diabetes dataset, even after training for 85 epochs, as reported in the original paper. Extending the training to 105 epochs does not rectify this issue, with 'max_glu_serum' and 'A1Cresult' being correctly generated in only approximately 2.8% and 12.1% of instances, respectively. For these columns, failure cases are treated as NaN values.

We utilize the official open-source code from the official TabDDPM GitHub repository to reproduce CTAB-GAN [39], CTAB-GAN+ [40], and TabDDPM [19]. For the Adult Income and California datasets, the hyperparameters provided are used. For other datasets lacking provided hyperparameters, we apply the hyperparameters from the Wilt dataset, which, according to the corresponding paper, demonstrated the largest performance improvement over the original data.

Due to the lack of official code provided by the authors of LITO [37], we adopt the prompt design described in their original paper. Specifically, we use the prompt for LITO-C to generate minority class data. To generate samples for both majority and minority classes, we also use the modified prompt to generate majority class data. Similarly, for CuratedLLM [27], we follow the prompt design detailed in their paper. Although LITO and CuratedLLM propose multi-stage techniques for data generation, we focus solely on evaluating their prompt designs.

### C.4 Experimental setup

For classification, we utilize the target class of each dataset to divide groups. Specifically, the Diabetes dataset has three groups, and the Travel, Sick, HELOC, Income, and Thyroid datasets have two groups. For the process of unique variable mapping, each discrete value is transformed into a combination of three characters, including uppercase letters and digits. Regarding hyperparameter settings of downstream machine learning models: For the gradient boosting classifier, we use the default hyperparameters provided by scikit-learn library. For XGBoost [7], CatBoost [24], and LightGBM [17], we conduct 5-fold cross-validation on the training set of the original data, optimizing two key hyperparameters: learning rate and max depth. These optimized settings are then consistently applied across all experiments, aligning the model's performance closely with the characteristics of the original data. Where possible, we train the classification models using a single NVIDIA GeForce RTX 3090 GPU on an Ubuntu 18.04 system.

### C.5 More details on the prompt design

This section outlines the construction of the proposed prompt. As shown in the prompt examples in Tables 13-18, the final prompt template for synthetic data generation consists of four parts.

**Descriptions (Optional).** The prompt may start with descriptions of the variables in the dataset, provided line-by-line, if precise and available. These descriptions define what each column in the dataset represents, offering contextual understanding to the LLM about the data. It will be located at the top of the prompt (but not repeated to maintain token efficiency) for additional context.

**Set-level examples.** Then, a set-level examples are present. This section begins with a data entry format header, specifying the order and names of each feature. This header acts as a guide for the LLM, indicating the start of an example set, which is crucial for interpreting the structure of the tabular data. Following the header, there's a line specifying a group name, signaling the start of a specific group. Instances belonging to this group are then listed, one per line, in a CSV style. This process is repeated for each group within the set.

**Repetition of set-Level template.** To reinforce the data structure and patterns, this set-level template is repeated several times within the prompt, each time with distinct data samples. This repetition is key in enabling the LLM to recognize, learn, and subsequently replicate the set format in its synthetic data generation process.

**Trigger for LLM data generation.** After all the example sets are presented, the same header that indicates the start of an example set is included at the end of the prompt. This header serves as a signal for the LLM to commence generating synthetic data, maintaining the structure and patterns established in the prompt.

By following this structured approach, the LLM is guided to effectively generate synthetic data that matches the structure and characteristics of the original dataset.

## D  Related work

**LLM in-context learning prompting**    With the remarkable advancements in LLMs, extensive research has been conducted in prompt engineering to maximize their potential for various natural

language processing (NLP) tasks, including text summarization and question answering, yielding significant value [9, 28, 36, 41]. Optimizing prompts for specific tasks is inherently a combinatorial challenge, and in the absence of established optimization principles, progress has often been driven by heuristic methods validated through rigorous empirical evaluations [25, 32, 38]. Our study adopts this empirical approach, conducting extensive experiments to develop effective prompts for synthetic tabular data generation, which holds unique challenges that differ from typical NLP tasks. Through these experiments, we introduce a tailored prompt design, EPIC, that employs class-balanced, grouped, and structured formatting and unique variable mapping.

**Tabular data generation** In the field of tabular data synthesis, traditional methods have predominantly focused on interpolation techniques, such as SMOTE [5]. Although useful, these methods often struggle to capture complex relationships between features [2]. Recently, generative models have shown promise in generating realistic tabular data [6, 8, 16, 35, 39, 40], and diffusion models have also achieved promising results in this area [19]. Building on these advances, GReaT [4] achieved notable success by transforming tabular data into natural text format and fine-tuning LLMs for synthetic tabular data generation. However, this fine-tuning process is resource-intensive, requiring extensive training of large models for each dataset. To address this, LITO [37] and Curated LLM [27] have explored using LLMs through in-context learning for tabular data generation, yet these approaches require additional filtering steps to ensure data quality. In contrast, we demonstrate that with comprehensive prompt exploration, high-quality tabular data generation can be achieved through prompt design alone, without the need for additional steps. Our work is distinguished by the thorough and comprehensive evaluations across six real-world datasets and synthetic toy data, aiming to provide deeper insights into the prompting process. We addressed experiments and analyses often overlooked in prior research on tabular data generation.

# E   Broader Impact

Our method is generally applicable to tabular data generation and classification tasks with minimal preprocessing, making it accessible for diverse research and industrial applications. Potential positive impacts include enhanced data accessibility and privacy preservation, which can support decision-making processes in fields such as healthcare and finance by reducing reliance on real data and protecting individual privacy. However, potential negative impacts include the risk of data misuse. Synthetic data generation could be exploited maliciously, such as to create deceptive datasets intended to mislead systems or individuals. Additionally, using closed-source LLMs poses potential risks, including data leakage through API calls, particularly concerning when handling sensitive data, and limitations on direct access to verify the models being used. Ensuring quality, security, and ethical use of generated data under these conditions is essential.

# F   Complete quantitative results

This section presents the complete confusion matrix results in Appendix F.1 and compares the classification performance with the baselines in Appendix F.2.

## F.1   Confusion matrix results

We present normalized confusion matrix results on binary classification datasets, as shown in Fig. 14. The results demonstrate that adding the synthetic data generated by our method to the original dataset significantly enhances classification performance, outperforming baselines by a large margin. Additionally, using only the synthetic data generated by our method achieves better performance than augmenting the original dataset with synthetic data from baselines.

## F.2   Complete results on comparison with baselines

Table 19 offers the complete performance comparison results with standard deviation values of 20 independent experiments on all six datasets.

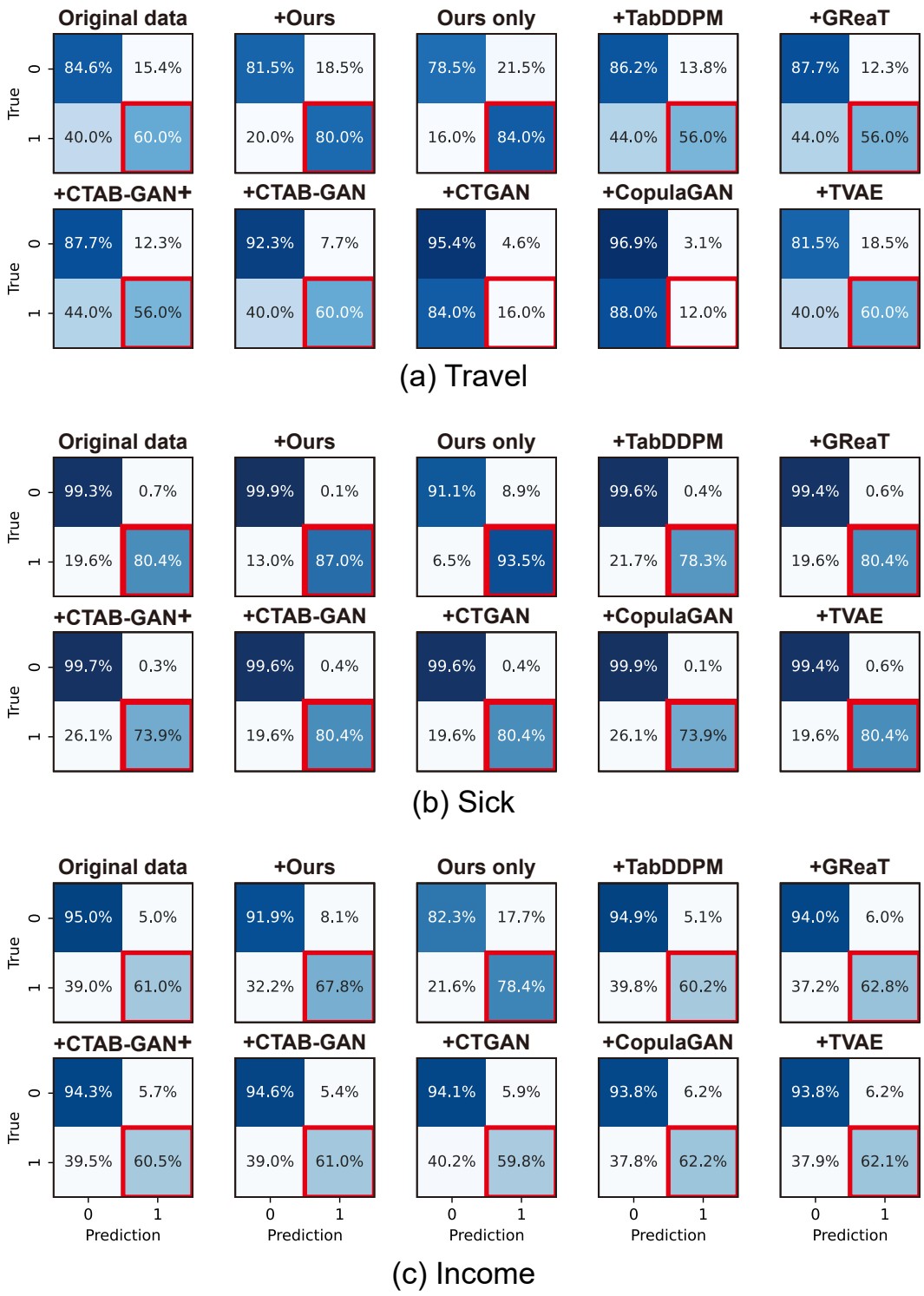

Figure 14: **Normalized confusion matrix results on binary classification datasets using the gradient boosting classifier.** `Ours only` denotes cases where only our synthetic data are used.

Table 13: **Example of an EPIC prompt for the Travel dataset.** This prompt illustrates the structure of sets and groups. Each iteration utilizes randomly selected samples from the training data.

| Template | | Prompt sample |
|---|---|---|
| Descriptions | | `Churn: whether customer churns or doesnt churn for tour and travels company,`
`age: the age of customer,`
`FrequentFlyer: whether customer takes frequent flights,`
`AnnualIncomeClass: class of annual income of user,`
`ServicesOpted: number of times services opted during recent years,`
`AccountSyncedToSocialMedia: whether company account of user synchronised to their social media,`
`BookedHotelOrNot: whether the customer book lodgings/Hotels using company services.\n\n` |
| Set | Header | `Churn,Age,FrequentFlyer,AnnualIncomeClass,ServicesOpted,`
`AccountSyncedToSocialMedia,BookedHotelOrNot` |
| | Group

Group | `A.`
`Churn,28,Yes,High Income,6,No,Yes`
`Churn,37,Yes,Low Income,4,Yes,Yes`
`Churn,30,Yes,Low Income,1,Yes,Yes\n`
`B.`
`Doesnt churn,38,No,Low Income,1,Yes,No`
`Doesnt churn,28,No Record,Low Income,5,No,Yes`
`Doesnt churn,34,Yes,Low Income,1,No,No\n\n` |
| Set | Header | `Churn,Age,FrequentFlyer,AnnualIncomeClass,ServicesOpted,`
`AccountSyncedToSocialMedia,BookedHotelOrNot` |
| | Group

Group | `A.`
`Churn,30,Yes,High Income,4,No,No`
`Churn,28,No,Low Income,6,No,Yes`
`Churn,28,No Record,Middle Income,2,No,No\n`
`B.`
`Doesnt churn,37,Yes,Low Income,1,No,No`
`Doesnt churn,36,No,Middle Income,1,No,Yes`
`Doesnt churn,28,No,Middle Income,3,No,No\n\n` |
| Set | Header | `Churn,Age,FrequentFlyer,AnnualIncomeClass,ServicesOpted,`
`AccountSyncedToSocialMedia,BookedHotelOrNot` |
| | Group

Group | `A.`
`Churn,27,Yes,High Income,5,No,No`
`Churn,37,No,Low Income,5,Yes,No`
`Churn,33,No,Low Income,5,Yes,Yes\n`
`B.`
`Doesnt churn,37,Yes,Low Income,1,No,No`
`Doesnt churn,36,No,Middle Income,1,No,Yes`
`Doesnt churn,28,No,Middle Income,3,No,No\n\n` |
| Set | Header
(*trigger*) | `Churn,Age,FrequentFlyer,AnnualIncomeClass,ServicesOpted,`
`AccountSyncedToSocialMedia,BookedHotelOrNot` |

Table 14: **Example of an EPIC prompt for the Sick dataset.** This prompt illustrates the structure of sets and groups. Each iteration utilizes randomly selected samples from the training data. In this scenario, our unique variable mapping is employed, whereby each unique value of a variable is consistently substituted with a unique three-character alphanumeric string. This approach ensures diversity and robustness in the synthesized data.

| Template | | Prompt sample |
|---|---|---|
| Descriptions | | `Class: hypothyroidism is a condition in which the thyroid gland is underperforming or producing too little thyroid hormone,` `age: the age of an patient,` `sex: the biological sex of an patient,` `TSH: thyroid stimulating hormone,` `T3: triiodothyronine hormone,` `TT4: total levothyroxine hormone,` `T4U: levothyroxine hormone uptake,` `FTI: free levothyroxine hormone index,` `referral_source: institution that supplied the thyroid disease record.\n\n` |
| Set | Header | `Class,age,sex,on_thyroxine,query_on_thyroxine,` `on_antithyroid_medication,sick,pregnant,thyroid_surgery,` `I131_treatment,query_hypothyroid,query_hyperthyroid,lithium,` `goitre,tumor,hypopituitary,psych,TSH_measured,TSH,T3_measured,T3,` `TT4_measured,TT4,T4U_measured,T4U,FTI_measured,FTI,referral_source` |
| | Group | `A.`
`JY0,64.0,E3R,ZIQ,A6A,K6Y,RU5,SQ6,Q6D,IER,Z9P,Z50,Y9J,SYD,ZWI,PDL,`
`UZ8,KWH,0.85,PIX,1.1,ASS,99.0,D0T,1.11,SD4,90.0,X5Z`
`JY0,72.0,L2J,TU1,A6A,K6Y,RU5,SQ6,Q6D,IER,TFG,Z50,Y9J,SYD,CLC,PDL,`
`UZ8,KWH,0.28,PIX,0.9,ASS,79.0,D0T,0.7,SD4,112.0,X5Z\n` |
| | Group | `B.`
`GGN,56.0,L2J,TU1,A6A,K6Y,RU5,SQ6,Q6D,IER,TFG,Z50,Y9J,SYD,ZWI,PDL,`
`UZ8,KWH,5.4,PIX,1.7,ASS,104.0,D0T,1.01,SD4,103.0,X5Z`
`GGN,42.0,OQX,TU1,A6A,K6Y,RU5,SQ6,Q6D,IER,TFG,Z50,Y9J,SYD,ZWI,PDL,`
`UZ8,KWH,0.02,PIX,2.6,ASS,138.0,D0T,1.58,SD4,88.0,W7B\n\n` |
| Set | Header | `Class,age,sex,on_thyroxine,query_on_thyroxine,` `on_antithyroid_medication,sick,pregnant,thyroid_surgery,` `I131_treatment,query_hypothyroid,query_hyperthyroid,lithium,` `goitre,tumor,hypopituitary,psych,TSH_measured,TSH,T3_measured,T3,` `TT4_measured,TT4,T4U_measured,T4U,FTI_measured,FTI,referral_source` |
| | Group | `A.`
`JY0,72.0,L2J,TU1,A6A,K6Y,RU5,SQ6,Q6D,IER,TFG,Z50,Y9J,SYD,ZWI,PDL,`
`UZ8,KWH,5.3,PIX,1.0,ASS,97.0,D0T,0.65,SD4,150.0,X5Z`
`JY0,60.0,L2J,TU1,A6A,K6Y,RU5,SQ6,Q6D,IER,TFG,Z50,Y9J,SYD,ZWI,PDL,`
`UZ8,KWH,1.2,PIX,0.8,ASS,44.0,D0T,0.84,SD4,52.0,X5Z\n` |
| | Group | `B.`
`GGN,23.0,E3R,TU1,A6A,K6Y,RU5,SQ6,Q6D,IER,TFG,Z50,Y9J,SYD,ZWI,PDL,`
`UZ8,KWH,3.6,PIX,7.0,ASS,141.0,D0T,1.77,SD4,80.0,YF8`
`GGN,32.0,E3R,TU1,A6A,K6Y,RU5,SQ6,Q6D,IER,TFG,Z50,Y9J,SYD,ZWI,PDL,`
`UZ8,KWH,0.64,PIX,1.7,ASS,102.0,D0T,0.76,SD4,134.0,YF8\n\n` |
| Set | Header (*trigger*) | `Class,age,sex,on_thyroxine,query_on_thyroxine,` `on_antithyroid_medication,sick,pregnant,thyroid_surgery,` `I131_treatment,query_hypothyroid,query_hyperthyroid,lithium,` `goitre,tumor,hypopituitary,psych,TSH_measured,TSH,T3_measured,T3,` `TT4_measured,TT4,T4U_measured,T4U,FTI_measured,FTI,referral_source` |

Table 15: **Example of an EPIC prompt for the HELOC dataset.** This prompt illustrates the structure of sets and groups. Each iteration utilizes randomly selected samples from the training data.

| Template | | Prompt sample |
|---|---|---|
| Descriptions | | |
| Set | Header | `RiskPerformance,ExternalRiskEstimate,MSinceOldestTradeOpen,` `MSinceMostRecentTradeOpen,AverageMInFile,NumSatisfactoryTrades,` `NumTrades60Ever2DerogPubRec,NumTrades90Ever2DerogPubRec,` `PercentTradesNeverDelq,MSinceMostRecentDelq,` `MaxDelq2PublicRecLast12M,MaxDelqEver,NumTotalTrades,` `NumTradesOpeninLast12M,PercentInstallTrades,` `MSinceMostRecentInqexcl7days,NumInqLast6M,NumInqLast6Mexcl7days,` `NetFractionRevolvingBurden,NetFractionInstallBurden,` `NumRevolvingTradesWBalance,NumInstallTradesWBalance,` `NumBank2NatlTradesWHighUtilization,PercentTradesWBalance` |
| | Group | `A.` Bad,90,211,6,102,17,0,0,100,-7,7,8,17,1,0,0,1,1,0,-8,0,-8,0,0 Bad,56,146,3,41,37,0,0,100,-7,7,8,41,4,24,4,1,1,75,75,15,2,5,90\n |
| | Group | `B.` Good,87,222,8,111,28,0,0,97,-8,6,6,33,1,24,0,0,0,0,13,2,2,0,27 Good,81,302,2,86,37,0,0,95,59,6,6,41,4,41,0,0,0,1,69,3,5,0,50\n\n |
| Set | Header | `RiskPerformance,ExternalRiskEstimate,MSinceOldestTradeOpen,` `MSinceMostRecentTradeOpen,AverageMInFile,NumSatisfactoryTrades,` `NumTrades60Ever2DerogPubRec,NumTrades90Ever2DerogPubRec,` `PercentTradesNeverDelq,MSinceMostRecentDelq,` `MaxDelq2PublicRecLast12M,MaxDelqEver,NumTotalTrades,` `NumTradesOpeninLast12M,PercentInstallTrades,` `MSinceMostRecentInqexcl7days,NumInqLast6M,NumInqLast6Mexcl7days,` `NetFractionRevolvingBurden,NetFractionInstallBurden,` `NumRevolvingTradesWBalance,NumInstallTradesWBalance,` `NumBank2NatlTradesWHighUtilization,PercentTradesWBalance` |
| | Group | `A.` Bad,63,150,3,50,29,3,3,91,75,6,3,32,4,75,-7,6,6,33,90,1,7,1,73 Bad,73,269,14,91,28,0,0,97,-8,6,6,34,0,26,0,0,0,49,70,5,4,2,75\n |
| | Group | `B.` Good,79,344,24,135,24,0,0,100,-7,7,8,24,0,17,1,1,1,41,-8,4,1,2,50 Good,85,386,2,125,25,0,0,96,34,6,6,49,1,39,1,1,1,2,72,4,3,0,64\n\n |
| Set | Header (*trigger*) | `RiskPerformance,ExternalRiskEstimate,MSinceOldestTradeOpen,` `MSinceMostRecentTradeOpen,AverageMInFile,NumSatisfactoryTrades,` `NumTrades60Ever2DerogPubRec,NumTrades90Ever2DerogPubRec,` `PercentTradesNeverDelq,MSinceMostRecentDelq,` `MaxDelq2PublicRecLast12M,MaxDelqEver,NumTotalTrades,` `NumTradesOpeninLast12M,PercentInstallTrades,` `MSinceMostRecentInqexcl7days,NumInqLast6M,NumInqLast6Mexcl7days,` `NetFractionRevolvingBurden,NetFractionInstallBurden,` `NumRevolvingTradesWBalance,NumInstallTradesWBalance,` `NumBank2NatlTradesWHighUtilization,PercentTradesWBalance` |

Table 16: **Example of an EPIC prompt for the Income dataset.** This prompt illustrates the structure of sets and groups. Each iteration utilizes randomly selected samples from the training data.

| Template | | Prompt sample |
|---|---|---|
| Descriptions | | |
| Set | Header | `income,age,workclass,fnlwgt,education,education_num,` `marital-status,occupation,relationship,race,sex,capital-gain,` `capital-loss,hours-per-week,native-country` |
| | Group | `A.`
>50K,52,Private,298215,Bachelors,13,Married-civ-spouse,
Craft-repair,Husband,White,Male,0,0,50,United-States
>50K,30,Self-emp-inc,321990,Masters,14,Married-civ-spouse,
Exec-managerial,Husband,White,Male,15024,0,60,?
>50K,28,Local-gov,33662,Masters,14,Married-civ-spouse,
Prof-specialty,Wife,White,Female,7298,0,40,United-States\n |
| | Group | `B.`
<=50K,42,Private,572751,Preschool,1,Married-civ-spouse,
Craft-repair,Husband,White,Male,0,0,40,Nicaragua
<=50K,25,Self-emp-not-inc,159909,Assoc-voc,11,Married-civ-spouse,
Farming-fishing,Husband,White,Male,0,0,40,United-States
<=50K,20,?,182117,Some-college,10,Never-married,?,Own-child,White,
Male,0,0,40,United-States\n\n |
| Set | Header | `income,age,workclass,fnlwgt,education,education_num,` `marital-status,occupation,relationship,race,sex,capital-gain,` `capital-loss,hours-per-week,native-country` |
| | Group | `A.`
>50K,38,Private,58108,Bachelors,13,Married-civ-spouse,
Exec-managerial,Husband,White,Male,0,0,50,United-States
>50K,31,State-gov,124020,Assoc-acdm,12,Married-civ-spouse,
Tech-support,Husband,White,Male,0,0,40,United-States
>50K,41,Private,130126,Prof-school,15,Married-civ-spouse,
Prof-specialty,Husband,White,Male,0,0,80,United-States\n |
| | Group | `B.`
<=50K,51,Private,138514,Assoc-voc,11,Divorced,Tech-support,
Unmarried,Black,Female,0,0,48,United-States
<=50K,18,Private,205218,11th,7,Never-married,Sales,Own-child,White,
Female,0,0,20,United-States
<=50K,21,Private,185948,Some-college,10,Never-married,Sales,
Own-child,White,Male,0,0,35,United-States\n\n |
| Set | Header (*trigger*) | `income,age,workclass,fnlwgt,education,education_num,` `marital-status,occupation,relationship,race,sex,capital-gain,` `capital-loss,hours-per-week,native-country` |

Table 17: **Example of an EPIC prompt for the Diabetes dataset.** This prompt illustrates the structure of sets and groups. Each iteration utilizes randomly selected samples from the training data. In this scenario, our unique variable mapping is employed, whereby each unique value of a variable is consistently substituted with a unique three-character alphanumeric string. This approach ensures diversity and robustness in the synthesized data.

| Template | | Prompt sample |
|---|---|---|
| Descriptions | | |
| Set | Header | `readmitted,encounter_id,patient_nbr,race,gender,age,weight,` `admission_type_id,discharge_disposition_id,admission_source_id,` `time_in_hospital,payer_code,medical_specialty,num_lab_procedures,` `num_procedures,num_medications,number_outpatient,number_emergency,` `number_inpatient,diag_1,diag_2,diag_3,number_diagnoses,` `max_glu_serum,A1Cresult,metformin,repaglinide,nateglinide,` `chlorpropamide,glimepiride,acetohexamide,glipizide,glyburide,` `tolbutamide,pioglitazone,rosiglitazone,acarbose,miglitol,` `troglitazone,tolazamide,insulin,glyburide-metformin,` `glipizide-metformin,glimepiride-pioglitazone,` `metformin-rosiglitazone,metformin-pioglitazone,change,diabetesMed` |
| | Group | `A.` 
 `AD6,39850434,428733,HX2,LL7,10F,H7H,2,7,1,13,QTD,3S2,44,4,15,1,0,` 
 `6,QDY,HR7,8K9,9,JAT,NOH,K06,2TV,65H,A7C,MK7,JWO,HXK,SVQ,BMY,FCV,` 
 `ZRU,IDJ,8CO,78A,NXL,CWD,6TQ,VDM,HMH,OOU,HNM,NGY,NC4` 
 `AD6,112757142,66907593,7MT,LL7,10F,BES,2,1,1,1,CFV,XY0,50,3,16,3,0,` 
 `3,6RA,BSI,DQ8,8,JAT,NOH,K06,2TV,65H,A7C,MK7,JWO,HXK,DQV,BMY,FCV,` 
 `ZRU,IDJ,8CO,78A,NXL,PDO,6TQ,VDM,HMH,OOU,HNM,UXM,NC4\n` |
| | Group | `B.` 
 `YRB,163732050,91571517,7MT,LL7,BM0,H7H,3,1,1,1,QTD,SKI,17,6,9,0,0,` 
 `1,6RA,4UR,OTR,9,JAT,NOH,K06,2TV,65H,A7C,MK7,JWO,HXK,SVQ,BMY,FCV,` 
 `ZRU,IDJ,8CO,78A,NXL,PDO,6TQ,VDM,HMH,OOU,HNM,UXM,R51` 
 `YRB,108763158,24232068,7MT,UWV,TVU,H7H,1,6,7,7,QBR,79B,46,0,17,0,0,` 
 `0,I3K,OOH,6BC,9,JAT,NOH,K06,2TV,65H,A7C,GJH,JWO,HXK,SVQ,BMY,FCV,` 
 `ZRU,IDJ,8CO,78A,NXL,05I,6TQ,VDM,HMH,OOU,HNM,NGY,NC4\n` |
| | Group | `C.` 
 `PS6,274193592,68737500,7MT,LL7,10F,H7H,1,3,7,5,QBR,SKI,60,1,18,0,0,` 
 `0,DL8,AL8,P2T,9,JAT,ZAX,K06,2TV,65H,A7C,MK7,JWO,HXK,SVQ,BMY,FCV,` 
 `ZRU,IDJ,8CO,78A,NXL,CWD,6TQ,VDM,HMH,OOU,HNM,NGY,NC4` 
 `PS6,156572340,114902370,7MT,UWV,BM0,H7H,1,1,7,2,QSB,SKI,50,3,8,0,0,` 
 `0,24G,FKO,DZJ,9,JAT,NOH,K06,2TV,65H,A7C,MK7,JWO,HXK,SVQ,BMY,FCV,` 
 `ZRU,IDJ,8CO,78A,NXL,8GG,6TQ,VDM,HMH,OOU,HNM,NGY,NC4\n\n` |
| | | . . . |
| Set | Header (*trigger*) | `readmitted,encounter_id,patient_nbr,race,gender,age,weight,` `admission_type_id,discharge_disposition_id,admission_source_id,` `time_in_hospital,payer_code,medical_specialty,num_lab_procedures,` `num_procedures,num_medications,number_outpatient,number_emergency,` `number_inpatient,diag_1,diag_2,diag_3,number_diagnoses,` `max_glu_serum,A1Cresult,metformin,repaglinide,nateglinide,` `chlorpropamide,glimepiride,acetohexamide,glipizide,glyburide,` `tolbutamide,pioglitazone,rosiglitazone,acarbose,miglitol,` `troglitazone,tolazamide,insulin,glyburide-metformin,` `glipizide-metformin,glimepiride-pioglitazone,` `metformin-rosiglitazone,metformin-pioglitazone,change,diabetesMed` |

Table 18: **Example of an EPIC prompt for the Thyroid dataset.** This prompt illustrates the structure of sets and groups. Each iteration utilizes randomly selected samples from the training data.

| Template | | Prompt sample |
|---|---|---|
| Descriptions | | |
| Set | Header | `Recurred,Age,Gender,Smoking,Hx Smoking,Hx Radiothreapy,` `Thyroid Function,Physical Examination,Adenopathy,Pathology,` `Focality,Risk,T,N,M,Stage,Response` |
| | Group | `A.`
`Yes,46,M,Yes,No,No,Euthyroid,Single nodular goiter-left,`
`Bilateral,Follicular,Uni-Focal,High,T4b,N1b,M1,II,Structural`
`Incomplete`
`Yes,27,M,No,No,No,Euthyroid,Multinodular goiter,Bilateral,`
`Papillary,Multi-Focal,Intermediate,T3a,N1b,M0,I, Structural`
`Incomplete`
`Yes,35,F,No,No,No,Euthyroid,Multinodular goiter,Right,`
`Papillary,Multi-Focal,Intermediate,T1b,N1b,M0,I, Structural`
`Incomplete\n` |
| | Group | `B.`
`No,31,M,No,No,No,Euthyroid,Single nodular goiter-right,No,`
`Papillary,Uni-Focal,Low,T3a,N0,M0,I,Indeterminate`
`No,25,F,No,No,No,Euthyroid,Multinodular goiter,No,`
`Papillary,Uni-Focal,Low,T2,N0,M0,I,Indeterminate`
`No,30,F,No,No,No,Euthyroid,Single nodular goiter-right,No,`
`Papillary,Uni-Focal,Low,T1b,N0,M0,I,Excellent\n` |
| Set | Header | `Recurred,Age,Gender,Smoking,Hx Smoking,Hx Radiothreapy,` `Thyroid Function,Physical Examination,Adenopathy,Pathology,` `Focality,Risk,T,N,M,Stage,Response` |
| | Group | `A.`
`Yes,37,M,No,No,No,Euthyroid,Multinodular goiter,Bilateral,`
`Papillary,Multi-Focal,Intermediate,T3a,N1b,M0,I,Structural`
`Incomplete`
`Yes,63,M,Yes,No,No,Euthyroid,Single nodular goiter-right,`
`Right,Papillary,Multi-Focal,Intermediate,T3a,N1b,M0,II,Structural`
`Incomplete`
`Yes,80,M,Yes,No,No,Euthyroid,Single nodular goiter-left,No,`
`Hurthel cell,Multi-Focal,Intermediate,T4a,N0,M0,II,Structural`
`Incomplete\n` |
| | Group | `B.`
`No,55,F,No,No,No,Euthyroid,Single nodular goiter-left,No,`
`Papillary,Uni-Focal,Low,T2,N0,M0,I,Excellent`
`No,31,F,No,No,No,Euthyroid,Multinodular goiter,Right,`
`Papillary,Multi-Focal,Intermediate,T1a,N1b,M0,I,Excellent`
`No,29,F,No,No,No,Euthyroid,Single nodular goiter-right,No,`
`Papillary,Uni-Focal,Low,T1b,N0,M0,I,Excellent\n` |
| Set | Header (*trigger*) | `Recurred,Age,Gender,Smoking,Hx Smoking,Hx Radiothreapy,` `Thyroid Function,Physical Examination,Adenopathy,Pathology,` `Focality,Risk,T,N,M,Stage,Response` |

Table 19: **Complete results comparing ML classification performance when synthetic data are added to the original dataset.** Results are averaged across four classifiers: XGBoost, CatBoost, LightGBM, and the gradient boosting classifier, with each model run five times. #syn denotes the number of synthetic samples added to the original dataset.

| Dataset | Method | #syn | F1 score ↑ | BAL ACC ↑ | Sensitivity ↑ | Specificity ↑ |
|---|---|---|---|---|---|---|
| Travel | Original | - | $58.12_{\pm2.04}$ (0.00) | $71.00_{\pm1.41}$ (0.00) | $57.00_{\pm3.40}$ (0.00) | $85.00_{\pm0.68}$ (0.00) |
| | +TVAE [35] | +1K | $59.78_{\pm4.89}$ (+1.66) | $72.35_{\pm3.52}$ (+1.35) | $62.00_{\pm6.81}$ (+5.00) | $82.69_{\pm1.31}$ (-2.31) |
| | +CopulaGAN [22] | +1K | $21.76_{\pm2.00}$ (-36.36) | $55.52_{\pm0.83}$ (-15.48) | $12.80_{\pm1.64}$ (-44.20) | $\textbf{98.23}_{\pm1.89}$ (+13.23) |
| | +CTGAN [35] | +1K | $29.84_{\pm3.61}$ (-28.28) | $57.79_{\pm1.56}$ (-13.21) | $19.20_{\pm2.46}$ (-37.80) | $96.38_{\pm1.15}$ (+11.38) |
| | +CTAB-GAN [39] | +1K | $56.07_{\pm8.29}$ (-2.05) | $69.58_{\pm5.14}$ (-1.42) | $51.00_{\pm7.88}$ (-6.00) | $88.15_{\pm3.31}$ (+3.15) |
| | +CTAB-GAN+ [40] | +1K | $54.66_{\pm3.64}$ (-3.46) | $68.62_{\pm2.40}$ (-2.38) | $53.00_{\pm3.40}$ (-4.00) | $84.23_{\pm2.05}$ (-0.77) |
| | +GReaT [4] | +1K | $60.95_{\pm2.59}$ (+2.83) | $72.86_{\pm1.80}$ (+1.86) | $58.80_{\pm3.69}$ (+1.80) | $86.92_{\pm0.79}$ (+1.92) |
| | +TabDDPM [19] | +1K | $53.20_{\pm4.10}$ (-4.92) | $67.70_{\pm2.69}$ (-3.30) | $50.40_{\pm4.19}$ (-6.60) | $85.00_{\pm1.31}$ (0.00) |
| | **+Ours** | +1K | $\textbf{66.65}_{\pm2.53}$ (+8.53) | $\textbf{78.23}_{\pm2.10}$ (+7.23) | $\textbf{78.00}_{\pm4.59}$ (+21.00) | $78.46_{\pm2.50}$ (-6.54) |
| Sick | Original | - | $87.81_{\pm2.46}$ (0.00) | $91.22_{\pm0.95}$ (0.00) | $82.83_{\pm1.71}$ (0.00) | $99.61_{\pm0.22}$ (0.00) |
| | +TVAE [35] | +1K | $87.77_{\pm2.88}$ (-0.04) | $91.47_{\pm1.33}$ (+0.25) | $83.37_{\pm2.47}$ (+0.54) | $99.56_{\pm0.23}$ (-0.05) |
| | +CopulaGAN [22] | +1K | $83.60_{\pm1.33}$ (-4.21) | $86.61_{\pm0.53}$ (-4.61) | $73.37_{\pm0.97}$ (-9.46) | $\textbf{99.86}_{\pm0.10}$ (+0.25) |
| | +CTGAN [35] | +1K | $87.52_{\pm1.68}$ (-0.29) | $89.86_{\pm0.52}$ (-1.36) | $79.89_{\pm0.97}$ (-2.94) | $99.82_{\pm0.19}$ (+0.21) |
| | +CTAB-GAN [39] | +1K | $86.12_{\pm1.49}$ (-1.69) | $89.51_{\pm1.24}$ (-1.71) | $79.35_{\pm2.49}$ (-3.48) | $99.68_{\pm0.06}$ (+0.07) |
| | +CTAB-GAN+ [40] | +1K | $82.35_{\pm4.16}$ (-5.46) | $86.28_{\pm2.49}$ (-4.94) | $72.83_{\pm4.86}$ (-10.00) | $99.74_{\pm0.14}$ (+0.13) |
| | +GReaT [4] | +1K | $87.23_{\pm1.87}$ (-0.58) | $90.83_{\pm1.09}$ (-0.39) | $82.07_{\pm2.10}$ (-0.76) | $99.60_{\pm0.12}$ (-0.01) |
| | +TabDDPM [19] | +1K | $85.17_{\pm2.07}$ (-2.64) | $89.30_{\pm1.18}$ (-1.92) | $79.02_{\pm2.26}$ (-3.81) | $99.57_{\pm0.10}$ (-0.04) |
| | **+Ours** | +1K | $\textbf{88.71}_{\pm1.98}$ (+0.90) | $\textbf{92.93}_{\pm0.91}$ (+1.71) | $\textbf{86.41}_{\pm1.85}$ (+3.58) | $99.44_{\pm0.27}$ (-0.17) |
| HELOC | Original | - | $71.01_{\pm0.47}$ (0.00) | $73.21_{\pm0.31}$ (0.00) | $67.89_{\pm0.82}$ (0.00) | $78.52_{\pm0.34}$ (0.00) |
| | +TVAE [35] | +1K | $71.12_{\pm0.32}$ (+0.11) | $73.25_{\pm0.33}$ (+0.04) | $68.15_{\pm0.21}$ (+0.26) | $78.34_{\pm0.48}$ (-0.18) |
| | +CopulaGAN [22] | +1K | $71.23_{\pm0.27}$ (+0.22) | $73.32_{\pm0.25}$ (+0.11) | $68.37_{\pm0.31}$ (+0.48) | $78.26_{\pm0.36}$ (-0.26) |
| | +CTGAN [35] | +1K | $70.82_{\pm0.24}$ (-0.19) | $73.06_{\pm0.27}$ (-0.15) | $67.60_{\pm0.25}$ (-0.29) | $78.52_{\pm0.55}$ (0.00) |
| | +CTAB-GAN [39] | +1K | $70.60_{\pm0.41}$ (-0.41) | $72.87_{\pm0.21}$ (-0.34) | $67.39_{\pm0.97}$ (-0.50) | $78.36_{\pm0.74}$ (-0.16) |
| | +CTAB-GAN+ [40] | +1K | $71.03_{\pm0.05}$ (+0.02) | $73.15_{\pm0.10}$ (-0.06) | $68.13_{\pm0.21}$ (+0.24) | $78.17_{\pm0.38}$ (-0.35) |
| | +GReaT [4] | +1K | $70.35_{\pm0.33}$ (-0.66) | $72.96_{\pm0.24}$ (-0.25) | $66.22_{\pm0.49}$ (-1.67) | $\textbf{79.70}_{\pm0.21}$ (+1.18) |
| | +TabDDPM [19] | +1K | $70.65_{\pm0.18}$ (-0.36) | $72.89_{\pm0.14}$ (-0.32) | $67.51_{\pm0.35}$ (-0.38) | $78.26_{\pm0.30}$ (-0.26) |
| | **+Ours** | +1K | $\textbf{71.92}_{\pm0.11}$ (+0.91) | $\textbf{73.66}_{\pm0.17}$ (+0.45) | $\textbf{69.96}_{\pm0.21}$ (+2.07) | $77.35_{\pm0.51}$ (-1.17) |
| Income | Original | - | $66.90_{\pm2.12}$ (0.00) | $76.45_{\pm1.48}$ (0.00) | $57.28_{\pm3.41}$ (0.00) | $\textbf{95.61}_{\pm0.46}$ (0.00) |
| | +TVAE [35] | +20K | $66.96_{\pm1.36}$ (+0.06) | $76.80_{\pm1.11}$ (+0.35) | $59.13_{\pm2.92}$ (+1.85) | $94.48_{\pm0.71}$ (-1.13) |
| | +CopulaGAN [22] | +20K | $66.75_{\pm1.72}$ (-0.15) | $76.73_{\pm1.33}$ (+0.28) | $59.16_{\pm3.41}$ (+1.88) | $94.29_{\pm0.81}$ (-1.32) |
| | +CTGAN [35] | +20K | $66.22_{\pm0.82}$ (-0.68) | $76.27_{\pm0.65}$ (-0.18) | $57.95_{\pm1.73}$ (+0.67) | $94.59_{\pm0.44}$ (-1.02) |
| | +CTAB-GAN [39] | +20K | $66.48_{\pm2.11}$ (-0.42) | $76.31_{\pm1.51}$ (-0.14) | $57.45_{\pm3.61}$ (+0.17) | $95.17_{\pm0.58}$ (-0.44) |
| | +CTAB-GAN+ [40] | +20K | $66.49_{\pm1.14}$ (-0.41) | $76.42_{\pm0.90}$ (-0.03) | $58.14_{\pm2.41}$ (+0.86) | $94.70_{\pm0.63}$ (-0.91) |
| | +GReaT [4] | +20K | $67.95_{\pm1.36}$ (+1.05) | $77.51_{\pm1.05}$ (+1.06) | $60.69_{\pm2.56}$ (+3.41) | $94.33_{\pm0.51}$ (-1.28) |
| | +TabDDPM [19] | +20K | $66.85_{\pm1.83}$ (-0.05) | $76.50_{\pm1.34}$ (+0.05) | $57.70_{\pm3.26}$ (+0.42) | $95.30_{\pm0.60}$ (-0.31) |
| | **+Ours** | +20K | $\textbf{69.16}_{\pm1.01}$ (+2.26) | $\textbf{79.15}_{\pm0.82}$ (+2.70) | $\textbf{66.45}_{\pm1.98}$ (+9.17) | $91.85_{\pm0.49}$ (-3.76) |
| Diabetes | Original | - | $54.87_{\pm1.37}$ (0.00) | $42.07_{\pm1.23}$ (0.00) | $60.00_{\pm0.64}$ (0.00) | $60.73_{\pm1.63}$ (0.00) |
| | +TVAE [35] | +10K | $54.79_{\pm1.40}$ (-0.08) | $41.96_{\pm1.24}$ (-0.11) | $59.96_{\pm0.66}$ (-0.04) | $60.71_{\pm1.65}$ (-0.02) |
| | +CopulaGAN [22] | +10K | $54.27_{\pm1.48}$ (-0.60) | $41.59_{\pm1.26}$ (-0.48) | $59.73_{\pm0.75}$ (-0.27) | $59.97_{\pm1.65}$ (-0.76) |
| | +CTGAN [35] | +10K | $54.72_{\pm1.13}$ (-0.15) | $41.92_{\pm1.02}$ (-0.15) | $59.86_{\pm0.53}$ (-0.14) | $60.63_{\pm1.32}$ (-0.10) |
| | +CTAB-GAN [39] | +10K | $54.22_{\pm1.28}$ (-0.65) | $41.53_{\pm1.09}$ (-0.54) | $59.73_{\pm0.59}$ (-0.27) | $59.91_{\pm1.46}$ (-0.82) |
| | +CTAB-GAN+ [40] | +10K | $54.24_{\pm1.16}$ (-0.63) | $41.52_{\pm1.01}$ (-0.55) | $59.63_{\pm0.57}$ (-0.37) | $60.01_{\pm1.31}$ (-0.72) |
| | +GReaT [4] | +10K | $54.78_{\pm1.31}$ (-0.09) | $41.98_{\pm1.17}$ (-0.09) | $59.98_{\pm0.60}$ (-0.02) | $60.61_{\pm1.55}$ (-0.12) |
| | +TabDDPM [19] | +10K | $54.64_{\pm1.50}$ (-0.23) | $41.83_{\pm1.28}$ (-0.24) | $59.91_{\pm0.75}$ (-0.09) | $60.55_{\pm1.78}$ (-0.18) |
| | **+Ours** | +10K | $\textbf{54.94}_{\pm1.43}$ (+0.07) | $\textbf{42.14}_{\pm1.29}$ (+0.07) | $\textbf{60.04}_{\pm0.66}$ (+0.04) | $\textbf{60.82}_{\pm1.68}$ (+0.09) |
| Thyroid | Original | - | $94.23_{\pm1.99}$ (0.00) | $95.08_{\pm1.60}$ (0.00) | $91.14_{\pm3.12}$ (0.00) | $99.02_{\pm1.01}$ (0.00) |
| | +TVAE [35] | +1K | $90.45_{\pm1.89}$ (-3.78) | $92.20_{\pm1.65}$ (-2.88) | $86.36_{\pm3.30}$ (-4.78) | $98.04_{\pm0.00}$ (-0.98) |
| | +CopulaGAN [22] | +1K | $86.73_{\pm3.99}$ (-7.50) | $88.71_{\pm2.91}$ (-6.37) | $78.41_{\pm5.08}$ (-12.73) | $99.02_{\pm1.01}$ (0.00) |
| | +CTGAN [35] | +1K | $76.40_{\pm6.58}$ (-17.83) | $81.14_{\pm4.55}$ (-13.94) | $62.27_{\pm9.10}$ (-28.87) | $\textbf{100.0}_{\pm0.00}$ (+0.98) |
| | +CTAB-GAN [39] | +1K | $53.49_{\pm3.71}$ (-40.74) | $68.30_{\pm1.73}$ (-26.78) | $36.59_{\pm3.45}$ (-54.55) | $\textbf{100.0}_{\pm0.00}$ (+0.98) |
| | +CTAB-GAN+ [40] | +1K | $27.46_{\pm7.80}$ (-66.77) | $58.07_{\pm2.60}$ (-37.01) | $16.14_{\pm5.21}$ (-75.00) | $\textbf{100.0}_{\pm0.00}$ (+0.98) |
| | +GReaT [4] | +1K | $91.31_{\pm1.61}$ (-2.92) | $92.46_{\pm0.99}$ (-2.62) | $85.91_{\pm1.40}$ (-5.23) | $99.02_{\pm1.01}$ (0.00) |
| | +TabDDPM [19] | +1K | $94.39_{\pm1.09}$ (+0.16) | $96.26_{\pm0.50}$ (+1.18) | $\textbf{95.45}_{\pm0.00}$ (+4.31) | $97.06_{\pm1.01}$ (-1.96) |
| | **+Ours** | +1K | $\textbf{94.80}_{\pm1.02}$ (+0.57) | $\textbf{96.39}_{\pm0.62}$ (+1.31) | $95.23_{\pm1.02}$ (+4.09) | $97.55_{\pm0.87}$ (-1.47) |

