# OpenReview forum: "EPIC: Effective Prompting for Imbalanced-Class Data Synthesis in Tabular Data Classification via Large Language Models"
_NeurIPS.cc/2024/Conference — NeurIPS 2024 poster_

### Official Review · Reviewer_kZFF · 2024-07-05

**Soundness:** 2
**Presentation:** 2
**Contribution:** 2
**Rating:** 5
**Confidence:** 3

**Summary:**

The paper proposes a method by which large language models are utilized to generate synthetic tabular data in order to mitigate class imbalances in existing datasets. The authors provide some tips which they discovered to result in more reliable data being generated, such as enforcing a CSV format. Their methods are validated on 6 real world datasets and one toy dataset.

**Strengths:**

1. the justifications for CSV formatting, class presentation, variable mapping, and task specification are all clear and make sense
2. the choice of experimental results is motivated sufficiently well
3. the provided results are extensive, including the plethora of ablation studies in the appendix

**Weaknesses:**

1. In Table 1, outside of the Travel dataset, the other 5 datasets do not show a significant increase in F1 score or balanced accuracy between the proposed method and the next closest baseline. Given this, it is not clear how to interpret the utility of the proposed method.
2. The motivation for why a large language model would be used to generate synthetic data for tabular settings, at scale, is a bit lacking. In particular, there needs to be discussion about the reasons for why the use of an LLM (with all of the computational and prompt-related overhead) would be preferable to existing data augmentation techniques for tabular data. In particular, it is not clear that the stated results provide enough evidence to persuasively demonstrate the utility of using LLMs for this purpose

**Questions:**

1. In Table 1, for the row "Ours" --- which LLM was used to generate the synthetic data? If it one of the gpt-3.5 models, then that should be stated up front so as to not increase confusion.

**Limitations:**

The limitations are addressed in Appendix G. However, they should be moved up to the main paper for full transparency. Additionally, the provided limitations are quite sparse. It would be helpful if the authors could think a bit more about the real-world usability of their method, as well as the drawbacks associated with using potentially closed-source large language models for their synthetic data generation purposes.

---

> ### Author Rebuttal · Authors · 2024-08-07
>
> > Q1. In Table 1, outside of the Travel dataset, the other 5 datasets do not show a significant increase in F1 score or balanced accuracy between the proposed method and the next closest baseline. Given this, it is not clear how to interpret the utility of the proposed method.
>
> A1. Please see the general response, titled “**General Response: 2**.”
> ***
>
> > Q2. The motivation for why a large language model would be used to generate synthetic data for tabular settings, at scale, is a bit lacking. In particular, there needs to be discussion about the reasons for why the use of an LLM (with all of the computational and prompt-related overhead) would be preferable to existing data augmentation techniques for tabular data. In particular, it is not clear that the stated results provide enough evidence to persuasively demonstrate the utility of using LLMs for this purpose
>
> A2. Please see the general response, titled “**General Response: 1**.”
>
> ***
> > Q3. In Table 1, for the row "Ours" --- which LLM was used to generate the synthetic data? If it one of the gpt-3.5 models, then that should be stated up front so as to not increase confusion.
>
> A3. We appreciate the reviewer's attention to detail. The results labeled 'Ours' in Table 1 were generated using the GPT3.5-turbo model. To enhance clarity and avoid any potential confusion, we will explicitly specify the model used in Table 1 in our paper.
>
> ***
> > Q4. The limitations are addressed in Appendix G. However, they should be moved up to the main paper for full transparency. Additionally, the provided limitations are quite sparse. It would be helpful if the authors could think a bit more about the real-world usability of their method, as well as the drawbacks associated with using potentially closed-source large language models for their synthetic data generation purposes.
>
> A4. We appreciate your suggestion and will address the limitations in the main manuscript for full transparency.
>
> Regarding the real-world usability of our method, our initial explanation may have been unclear. When the training dataset is large and cannot be fully included in the LLM prompt due to token size limitations, only a subset can be used as examples for generating samples. If these prompt samples do not fully represent the original data distribution, the generated data may be incomplete and of low quality. As illustrated in Fig. 10, an LLM can only generate a half-circle when prompted with one, highlighting its inability to produce data beyond what is presented in the input.
>
> To address this limitation, our method employs multiple rounds of random sampling with replacement to create a combined dataset that more accurately represents the original distribution, resulting in improved machine learning classification performance. However, this approach still carries the risk that the samples may not fully capture the original data distribution. Future research could focus on developing techniques to better identify and sample key examples that more accurately reflect the entire dataset.
>
> The selection of input examples can either reinforce or mitigate existing data biases. We proposed methods such as balancing and grouping to provide examples aimed at reducing these biases. While our primary focus was on addressing class imbalance, this approach could be extended to feature grouping, allowing for the generation of less biased data across specific features.
>
> Additionally, we would like to clarify that our method has been tested with both open-source LLMs (Llama2 and Mistral) and closed-source LLMs (GPT3.5-turbo). The underlying code, data, and algorithms of the open-source LLMs are publicly available and can be accessed, reviewed, and modified by the research community. Notably, Mistral generated data with feature correlations more closely aligned with the original data than GPT3.5-turbo, particularly for minor classes, as shown in Figs. 4 and 5.
>
> **Our method is model-agnostic and can be effectively used with open-source LLMs, not just closed-source ones, thereby not being limited by the drawbacks associated with closed-source LLMs.** However, when using closed-source LLMs, it is important to consider potential risks such as data leakage via API calls when handling privacy-sensitive data and the inability to directly access and verify the model being used. Ensuring the quality and security of the generated data in these circumstances is crucial.

---

> > ### Comment · Reviewer_kZFF · 2024-08-12
> >
> > Thank you to the authors for their detailed feedback and responses. I have updated my review accordingly.

---

> > > ### Author Response · Authors · 2024-08-13
> > >
> > > Thank you for taking the time to review our work and for your valuable feedback. We appreciate your consideration and are pleased that our responses have been helpful in updating your review.
> > >
> > > We are committed to making further improvements and are prepared to address any additional suggestions or clarifications that could strengthen our work.

---

### Official Review · Reviewer_8bof · 2024-07-10

**Soundness:** 3
**Presentation:** 2
**Contribution:** 2
**Rating:** 5
**Confidence:** 4

**Summary:**

This paper proposed an LLM-driven tabular data class balancing approach, adapting the in-context learning paradigm and trying various formats and templates to explore optimal prompts to mitigate imbalanced tabular data. Experimental results showed that the proposed method alleviated the CSV-format data imbalance.

**Strengths:**

1. The authors probed CSV-format data generation from different perspectives, including data format, class presentation, variable mapping, and task specification.
2. Experiments and data analysis of this paper were adequate for understanding the proposed data balancing approach, and data performance has been improved to some degree.

**Weaknesses:**

1. The research purpose of this paper covers merely the CSV-format data, limiting the further adaptivity and impact of the proposed method.
2. This paper merely tried to find plausible optimal prompts by handcraft, lacking theory analysis, making the prompting even a black box.
3. Empirical results are not consistent on all test data, especially in the specificity aspect, indicating the instability of the prompting approach.
4. According to the prompting design, it seems that the prompting process is a bit sophisticated, however, there are no computing cost comparison experiments.

**Questions:**

1. This paper proposed to mitigate the label imbalance issue for tabular data by generating scarce label data, but how to guarantee the generated data and labels were matched?
2. The presented method is basically an instruction prompting approach, and why didn’t the authors analyze the LLM during the prompting process to make it more traceable?
3. How did the proposed method handle missing data or noisy features in the original dataset?
4. Empirical results on the Travel data outperform other data, does that mean that such a prompt is more suitable for it and there exists better prompts for other datasets?
5. The overall performance of the proposed method achieved limited improvement, and have the authors conducted the improvement-computing cost ratio to make the profit more intuitive?

**Limitations:**

The authors have adequately discussed the limitations.

---

> ### Author Rebuttal · Authors · 2024-08-07
>
> Due to space constraints, we denote weaknesses and questions as W and Q, respectively.
> > W1. Research scope
>
> A. Focusing on tabular data does not limit the adaptivity or impact of our method. Tabular data, composed of mixed variable types such as numerical and categorical variables, represents a widely applicable and essential data format. It is the backbone of many datasets in various fields, including finance, healthcare, manufacturing, and natural sciences [1]. Enhancing performance on tabular data has a significant impact on real-world applications.
>
> Substantial research has been dedicated to tabular data, such as data generation and classification [2]. By addressing class imbalance and enhancing classification outcomes, our work contributes to the advancement of this crucial field, significantly impacting various domains.
> ***
> >W2, Q2. LLM analysis
>
> A. With the remarkable advancements in LLMs, extensive research has been conducted in prompt engineering to maximize their potential, yielding significant value. Optimizing prompts for specific tasks is inherently a combinatorial challenge, and in the absence of established optimization principles, progress has often been driven by heuristic methods validated through rigorous empirical evaluations [3].
>
> Our study adopts this empirical approach, conducting extensive experiments to develop effective prompts for synthetic tabular data generation. **Our work is distinguished by the thorough and comprehensive evaluations we performed across six real-world datasets from various domains and synthetic toy data, aiming to provide deeper insights into the prompting process.** We addressed experiments and analyses often overlooked in prior research on tabular data generation:
> * We used a synthetic dataset to observe how different prompts influence the accuracy of generated data distributions (Figs. 1, 7). For example, multi-class prompts yielded more accurate distributions than single-class prompts, as they allow LLMs to contrast different classes (Fig. 1, 4th row).
> * We explored the sampling of input examples and their corresponding outputs in LLMs using the toy set, providing insights into the variability and reliability of generated data (Figs. 8, 9, 10).
> * We conducted three unique experiments to analyze the utility of our method in enhancing ML classification performance: augmenting the original dataset with generated data (Tables 1, 17), augmenting only the minority class similar to SMOTE (Table 6), and using only generated data (Table 7).
> * We distinctively analyzed feature correlations by separately examining minor and major classes, comparing them across all prompt variations (Figs. 4, 5).
> * We investigated how varying numbers of generated samples affect classification performance (Figs. 6, C).
> * We ablated prompt elements to compare classification performance (Tables 3, 5) and conducted unique analyses of token usage and LLM generation efficiency (Tables 4, 8, 9).
> * We conducted experiments using three LLMs: Mistral, Llama2, and gpt3.5-turbo (Table 2, Figs. 4, 5).
>
> These analyses demonstrate how specific prompt design components impact the quality of generated data, addressing concerns about prompt optimization being a black box. While our study provides substantial empirical insights, we acknowledge that some areas may need further exploration and welcome detailed feedback, with additional analyses available.
> ***
> >W3. Performance
>
> A. Please see General Response 2.
> ***
> >W4. Cost
>
> A. We compared the computational cost of our method and its ablated versions in Tables 4, 8, and 9. Given a fixed number of input samples, we evaluated (1) the number of input tokens required, (2) the number of valid generated samples, and (3) the generation success rate. Our proposed prompt demonstrates superior efficiency compared to other prompt designs.
> ***
> >Q1. Label matching
>
> A. We demonstrated that the data generated by our method aligns well with the labels through detailed experiments:
> * Our method successfully produces data samples that match the classes, as denoted by color in Fig. 1, even when compared to fine-tuned GReaT.
> * For a machine learning model to perform well in classification, the correlation between input data and labels in the training data must be precise. Adding our synthetic data to the original data consistently improved the F1 score and balanced accuracy across six datasets  (Table 1).
> * Our synthetic data exhibits the closest feature correlation with the original data for each class, outperforming the baselines. (Figs 4, 5).
>
> These results indicate that our method generates data that is more accurately matched to the classes than the baselines.
> ***
> >Q3. Preprocessing
>
> A. Similar to GReaT, we retained the original data, including missing or noisy features, leveraging the capabilities of LLMs. The exception was the Sick dataset, where we followed the source’s method, replacing categorical missing values with "?" and imputing numerical values with the mean.
> ***
> >Q4. Optimal prompt
>
> A. We carefully selected key prompt design choices and validated our method through extensive ablation studies across **multiple** datasets from diverse domains, including social, healthcare, and marketing:
> * ML classification performance on three datasets (Tables 3, 5)
> * Feature correlations on two datasets (Figs. 4, 5),
> * Generated data distribution on the toy set (Figs. 1, 7)
>
> While it is possible that more advanced prompts could be developed in future work, our approach currently represents the state-of-the-art method for tabular data generation.
> ***
> >Q5. Cost-benefit analysis
>
> A. Please see General Response 1.
> ***
> [1] Breugel et al., Why Tabular Foundation Models Should Be a Research Priority, ICML’24
> [2] Fang et al., Large Language Models (LLMs) on Tabular Data: Prediction, Generation, and Understanding - A Survey, TMLR’24
> [3] Sahoo, et al., A systematic survey of prompt engineering in large language models: Techniques and applications., arXiv’24

---

> ### Comment · Reviewer_8bof · 2024-08-12
>
> Thank you for the responses, some concerns were addressed, however, there are still some questions.
>
> 1. Since the proposed approach is a prompt-based generation, have the authors tried to optimize the fixed prompt with some prompt optimizing methods, such as the Self-discover [1] and OPRO [2]?
> 2. There was no detailed introduction for various tabular datasets and no case study, which made me confused about the generalizability of the proposed method.
> 3. For the label matching of generated data, the authors need to employ advanced tabular data classification algorithms to evaluate the quality.
>
> [1]. Self-Discover: Large Language Models Self-Compose Reasoning Structures.
>
> [2]. Large Language Models as Optimizers.

---

> ### Author Response · Authors · 2024-08-13
>
> Thank you for your valuable comments and for taking the time to review our paper. We are glad that some of your concerns have been addressed, and we appreciate the opportunity to clarify and expand on the additional points.
> ***
> > Q1. Prompt optimization
>
> A1. We appreciate the suggestion, but there are significant differences between the typical applications of these methods and the challenges of synthetic tabular data generation.
> Self-discover and OPRO are designed to enhance LLM accuracy in structured tasks like multiple-choice questions or math problems, where correctness is relatively explicit. However, synthetic tabular data cannot be directly evaluated for correctness, making these methods less applicable to our domain.
>
> Recognizing the potential, we adapted OPRO, which provides official code, to our task by optimizing prompts based on the classification accuracy of a robust classifier, CatBoost. Using the Thyroid dataset with GPT-3.5-turbo, we applied OPRO to optimize our prompt. Unfortunately, **the optimized prompts failed to produce valid synthetic tabular data**. For instance, one generated prompt was:
>
> `No,45,F,No,No,No,Euthyroid,Single nodular goiter-right,No,Papillary,Multi-Focal,Intermediate,T3a,N1b,M0,II,Excellent`
>
> This is because OPRO is designed to optimize simple instructions, e.g., `Take a deep breath and work on this problem step-by-step` or `Break this down`, as shown in Table 1 of the OPRO paper. **While effective for question-answering and reasoning tasks, these types of prompts are unsuitable for constructing the complex structures required in synthetic tabular data generation.**
>
> Our experiments indicate that our proposed approach remains the most effective method for generating high-quality synthetic tabular data. Future research could explore prompt optimization methods specifically designed for tabular data generation to further enhance performance.
> ***
> > Q2. Dataset details & Generalizability
>
> A2. As noted in Section 3, we provided dataset details in Appendix I.2. However, **we acknowledge that this information may not be sufficiently highlighted in the main text**.
>
> To address this, **we will revise the manuscript to include a comprehensive introduction to these datasets within the main text, clearly demonstrating the broad applicability and generalizability of our method.** We will also emphasize the importance of tabular data research in enhancing decision-making and efficiency in various real-world applications, better communicating the impact and significance of our work in multiple fields.
> ***
> > Q3. Advanced classifier for label matching
>
> A3. We employed top-performing tabular classifiers, XGBoost, CatBoost, LightGBM, and Gradient boosting classifier, known for their strong performance, often surpassing recent deep learning models. These models have served as robust baselines in tabular classification (TabR, ICLR’24 [1]) and have been used to assess generated data quality in tabular generation studies (the benchmark paper, NeurIPS’23 [2] and TabDDPM, ICML’23). In our work, we evaluated label matching quality by averaging results across 20 runs with these models, five runs each, demonstrating the superiority of our method.
>
> We conducted preliminary experiments using recent in-context learning-based tabular classification methods, TabPFN (ICLR’23) [3] and T-Table (KDD’24) [4], on Travel, as detailed in **Table B of the attached PDF**. We also tested TabR (ICLR’24) [1], an advanced deep-learning tabular classification model, using its official code.
> |Model|Original|+Ours|+TabDDPM|+GReaT|
> |-|-|-|-|-|
> |TabR|46.41|**60.78**|44.88|32.41|
>
> The F1 score results indicate that **while the advanced models like TabPFN, T-Table, and TabR did not outperform traditional classifiers, adding synthetic data generated by our method consistently led to significant performance improvements, even with advanced tabular classification models.** In contrast, baselines resulted in performance decreases. Our method uniquely and consistently enhanced the performance of different classifiers, demonstrating superior label matching quality.
>
> Furthermore, as discussed earlier, our label matching quality is validated, regardless of the classifier, by:
> * Distinct class distribution in the generated data (Fig. 1, Toy set)
> * Feature correlation similarity with the original data across classes (Fig. 4, Travel & Fig. 5, Sick)
>
> These findings affirm that our model produces data with the best label matching compared to other baselines.
> ***
> [1] TabR: Tabular Deep Learning Meets Nearest Neighbors, ICLR’24
> [2] Reimagining Synthetic Tabular Data Generation through Data-Centric AI: A Comprehensive Benchmark, NeurIPS’23
> [3] TabPFN: A Transformer That Solves Small Tabular Classification Problems in a Second, ICLR’23
> [4] From Supervised to Generative: A Novel Paradigm for Tabular Deep Learning with Large Language Models, KDD’24
> [5] Self-consistency improves chain of thought reasoning in language models, ICLR’23

---

> > ### Comment · Reviewer_8bof · 2024-08-14
> >
> > Thank you for the reply, my concerns were addressed, and I will update the rating score correspondingly.

---

> > > ### Author Response · Authors · 2024-08-14
> > >
> > > We are pleased that our clarifications addressed your concerns, and we appreciate your decision to raise the score. Thank you for your insightful feedback and for taking the time to review our work.
> > >
> > > We remain committed to addressing any further suggestions or clarifications that could strengthen our work.

---

### Official Review · Reviewer_p2sf · 2024-07-11

**Soundness:** 4
**Presentation:** 4
**Contribution:** 3
**Rating:** 8
**Confidence:** 4

**Summary:**

The paper explores the domain of tabular data generation using in-context learning with LLM, in order to improve performance of an ML classifier, especially in imbalanced classes scenarios. The paper explores different prompting techniques, with detailed results on 6 datasets as well as a visualized study on a toy dataset. The paper also compares the result performance boost on several ML classifiers, which is indeed significant. There's a further discussion of efficiency and stability of the generation itself, and how it is affected given the different prompting methods proposed. Results using three different LLMs are presented.

**Strengths:**

The paper is very clear and discusses almost all important points soundly. It can be used as a guide both for researches facing the problem of generating data for imbalanced class learning, and for researches who are planning to study on LLM prompting techniques in a methodical way. It also presents clear benefits to using the proposed method. I enjoyed reading this paper very much.

**Weaknesses:**

The main discussion that I missed in the paper was regarding - how much extra data ends up being generated, and does generating more and more data using this method make any sense? I understand that the method uses actual data points, and samples from the original dataset without replacement. This probably creates a limitation to data creation - a point which is not discussed in the main paper body, and should be added. Furthermore, it might make sense to scramble and re-sample the dataset to create even more samples. I would have liked to see a discussion of this point, both to how much the datasets are actually enlarged using the no-replacement sampling, and also if it makes sense to generate more data with replacement, with the limits of this method discussed - when do we reach repetition and duplication, without any further improvement to the classifiers.
I'm also missing the original dataset sizes and minority vs majority class distributions - to show more strongly the class imbalance. In fact, no dataset descriptions are given at all, which in my opinion takes away from the paper.

**Questions:**

I'd add which model was used in table 1 (I'm assuming it's ChatGPT).
I would stress better the counts of additional synthetic data and why do you sometimes use 1K and sometimes 10K.

**Limitations:**

Limitations are discussed in the appendix, which is okay. I would perhaps add further discussion on social implications, clarifying the meaning of fig 10 - this method might create more bias in already biased data.

---

> ### Author Rebuttal · Authors · 2024-08-07
>
> > Q1. The main discussion that I missed in the paper was regarding - how much extra data ends up being generated, and does generating more and more data using this method make any sense? I understand that the method uses actual data points, and samples from the original dataset without replacement. This probably creates a limitation to data creation - a point which is not discussed in the main paper body, and should be added. Furthermore, it might make sense to scramble and re-sample the dataset to create even more samples. I would have liked to see a discussion of this point, both to how much the datasets are actually enlarged using the no-replacement sampling, and also if it makes sense to generate more data with replacement, with the limits of this method discussed - when do we reach repetition and duplication, without any further improvement to the classifiers.
>
> A1. Thank you for your insightful comment. You are correct that sampling without replacement limits the amount of synthetic data that can be generated to the number of actual data points. However, we would like to clarify that our method uses sampling with replacement to generate synthetic data. While we ensure that there are no overlapping examples within each prompt to maintain diversity, each prompt is constructed using sampling with replacement. We will clearly state this in our main manuscript.
>
> In response to your suggestion, we conducted **experiments to evaluate how much the datasets can be enlarged and how this impacts classification performance, comparing sampling with and without replacement**. As shown in **Fig. C of the attached PDF**, performance improves steadily in both scenarios as the volume of generated data increases. However, the improvement is constrained when sampling without replacement due to the limited number of possible samples.
>
> When sampling with replacement, as the dataset size expands, there is a noticeable improvement in the balance between sensitivity and specificity, which contributes to enhanced overall performance, including gains in balanced accuracy (BAL ACC) and F1 score. Generating up to 40K synthetic data points resulted in even better performance than the 20K synthetic data points reported in Table 1 of our main manuscript. We also observed that as the volume of generated data continues to increase, the gains in balanced accuracy and F1 score eventually plateau, indicating diminishing returns and suggesting that further data generation beyond a certain point offers limited additional benefit.
>
> ***
> > Q2. I'm also missing the original dataset sizes and minority vs majority class distributions - to show more strongly the class imbalance. In fact, no dataset descriptions are given at all, which in my opinion takes away from the paper.
>
> A2. We appreciate your observation. The dataset descriptions, including the original dataset sizes, are provided in Table 10 of Appendix I.2. However, we recognize that we omitted the class distribution information, which is now included in **Fig. D of the attached PDF**. Notably, the datasets where our method showed the most significant performance improvements (Travel, Income, and Sick) exhibit substantial class imbalances. We will incorporate this information into the main paper to enhance the clarity and impact of our findings.
>
> ***
> > Q3. I'd add which model was used in table 1 (I'm assuming it's ChatGPT).
>
> A3. We appreciate the reviewer's attention to detail. The results labeled 'Ours' in Table 1 were generated using the GPT3.5-turbo model. To enhance clarity and avoid any potential confusion, we will explicitly specify the model used in Table 1 in our paper.
>
> ***
> > Q4. I would stress better the counts of additional synthetic data and why do you sometimes use 1K and sometimes 10K.
>
> A4. The number of synthetic data points we generated was based on the size of the original datasets. For datasets with fewer than 10K samples, we generated 1K synthetic data points; for larger datasets, we generated 10K.
>
> ***
> > Q5. Limitations are discussed in the appendix, which is okay. I would perhaps add further discussion on social implications, clarifying the meaning of fig 10 - this method might create more bias in already biased data.
>
> A5. Our initial explanation may have been unclear. When the training dataset is large and cannot be fully included in the LLM prompt due to token size limitations, only a subset can be used as examples for generating samples. If these prompt samples do not fully represent the original data distribution, the generated data may be incomplete and of low quality. As illustrated in Fig. 10, an LLM can only generate a half-circle when prompted with one, highlighting its inability to produce data beyond what is presented in the input.
>
> To address this limitation, our method employs multiple rounds of random sampling with replacement to create a combined dataset that more accurately represents the original distribution, resulting in improved machine learning classification performance. However, this approach still carries the risk that the samples may not fully capture the original data distribution. Future research could focus on developing techniques to better identify and sample key examples that more accurately reflect the entire dataset.
>
> The selection of input examples can either reinforce or mitigate existing data biases. We proposed methods such as balancing and grouping to provide examples aimed at reducing these biases. While our primary focus was on addressing class imbalance, this approach could be extended to feature grouping, allowing for the generation of less biased data across specific features.
>
> Additionally, in Appendix F, we discuss the social implications, particularly the potential misuse of generated data to deceive systems or individuals. This highlights the importance of ethical considerations in applying our methods.

---

### Official Review · Reviewer_zoZq · 2024-07-14

**Soundness:** 3
**Presentation:** 3
**Contribution:** 2
**Rating:** 6
**Confidence:** 3

**Summary:**

The paper investigates how to use LLMs to generate synthetic tabular data for mitigating class imbalance in machine learning tasks. By exploring various prompting methods, the authors aim to identify key design elements that optimize the generation performance. The paper shows that using GPT3.5/Mistral/LLaMA to balancing classes, and employing unique variable mapping produces realistic and reliable data, enhancing classification performance (XGBoost etc) for minor classes in imbalanced datasets.

**Strengths:**

1. The paper provides a detailed exploration of various prompt design elements, such as data format, class presentation, and variable mapping, offering valuable insights for future research in this direction.
2. The proposed methods are easy to implement and require minimal preprocessing, making them accessible for a wide range of applications in tabular data classification tasks.
3. The experimental results show that the proposed approach consistently improves machine learning classification performance, particularly for minor classes.

**Weaknesses:**

1. Why not use the LLM itself to perform classification: The designed method uses LLMs to generate data examples for imbalanced classes, which means the LLMs must already have a good ability in modeling the data distribution even for the imbalanced classes. If that's the case, why not use the LLM itself to perform classification directly with the help of in-context learning? I expect that it will be a strong baseline compared to the XGBoost classifiers. The only downside of LLM-based classifiers may be the inference cost, but we should include this baseline.

**Questions:**

1. Could you provide the experiment results of in-context learning LLM-based classifiers on these tasks?

**Limitations:**

Yes in Appendix G

---

> ### Author Rebuttal · Authors · 2024-08-07
>
> > Q1. Why not use the LLM itself to perform classification: The designed method uses LLMs to generate data examples for imbalanced classes, which means the LLMs must already have a good ability in modeling the data distribution even for the imbalanced classes. If that's the case, why not use the LLM itself to perform classification directly with the help of in-context learning? I expect that it will be a strong baseline compared to the XGBoost classifiers. The only downside of LLM-based classifiers may be the inference cost, but we should include this baseline.
>
> > Q2. Could you provide the experiment results of in-context learning LLM-based classifiers on these tasks?
>
> A1. Thank you for your insightful suggestion. We concur that the demonstrated ability of LLMs to generate high-quality synthetic data for imbalanced classes suggests their potential for directly addressing classification tasks. Indeed, this represents a promising avenue for future exploration.
>
> However, the challenge lies in **designing prompts to fully harness this potential.** The effectiveness of LLMs is heavily influenced by how the prompts are crafted. Numerous studies have shown that performance can vary significantly depending on prompt design, leading to ongoing research aimed at optimizing prompts for specific tasks [1,2]. Our work focuses on designing prompts that enable LLMs to efficiently generate high-quality tabular data, particularly to address class imbalance. While leveraging LLMs for direct classification is an intriguing direction to improve classification performance, it is beyond the scope of our present study.
>
> We conducted **preliminary experiments using existing in-context learning-based tabular data classification methods**, such as TabPFN [3], which utilizes a pretrained transformer, and T-Table [4], which employs LLMs. Specifically, as detailed in **Table B of the attached PDF**, we evaluated classification performance on the Travel dataset using the original data and synthetic data generated by tabular data generation methods. For T-Table, we employed the GPT3.5-turbo model and, due to token limits, included all the original data but only 200 synthetic samples within each input prompt. Additionally, we used voting across five inferences to enhance performance [5].
>
> The results indicate that while the newly introduced models did not outperform traditional classifiers, adding synthetic data generated by our method to the original data consistently led to the greatest performance improvements in both TabPFN and T-Table models. These findings underscore **the value of high-quality synthetic data in enhancing classifier performance across diverse models**, highlighting the importance of research on data generation as a distinct research area, separate from classifier development.
>
> Tabular data generation is a critical area of research with significant implications [6,7,8]. This task serves two primary purposes:
> * Enhancing classification performance in a model-agnostic manner through data augmentation, similar to SMOTE, as the results demonstrated in Tables 1, 2, and 6.
> * Generating synthetic data to replace original data in security or privacy-sensitive contexts, as the results demonstrated in Table 7.
>
> In fields such as healthcare, where obtaining new samples or achieving balanced class labels is challenging and where data may be noisy or incomplete, generating high-quality synthetic data is crucial.
>
> In conclusion, even as more advanced classification models for tabular data are developed in the future, **our proposed method could continue to play a crucial role in enhancing performance by generating high-quality synthetic data**, thereby potentially making a significant impact on the tabular data classification community and related tasks.
>
> ***
>
>
> [1] Kojima, et al., Large language models are zero-shot reasoners., NeurIPS’22
> [2] Sahoo, et al., A systematic survey of prompt engineering in large language models: Techniques and applications., arXiv’24
> [3] Hollmann, et al., TabPFN: A Transformer That Solves Small Tabular Classification Problems in a Second., ICLR’23
> [4] Wen, et al., From Supervised to Generative: A Novel Paradigm for Tabular Deep Learning with Large Language Models., KDD’24
> [5] Wang, et al., Self-consistency improves chain of thought reasoning in language models., ICLR’23
> [6] Yang, et al., Language-Interfaced Tabular Oversampling via Progressive Imputation and Self-Authentication. ICLR’24
> [7] Seedat, et al., Curated llm: Synergy of llms and data curation for tabular augmentation in ultra low-data regimes., ICML’24
> [8] Borisov, et al., Language models are realistic tabular data generators., ICLR’23

---

> > ### Comment · Reviewer_zoZq · 2024-08-13
> >
> > Thanks for the reply! The response resolved my concerns. I will raise my score.

---

> > > ### Author Response · Authors · 2024-08-13
> > >
> > > Thank you for your valuable feedback and for taking the time to review our work. We are glad our clarifications addressed your concerns, and we appreciate your consideration in raising the score.
> > >
> > > We remain committed to addressing any further suggestions or clarifications that could enhance the quality of our work.

---

### Author Rebuttal · Authors · 2024-08-07

We sincerely appreciate the reviewers' valuable feedback and positive support, recognizing our method as
* Providing valuable guidance for researchers addressing class imbalance (zoZq, p2sf).
* Clear, well-articulated, and comprehensive in covering the method (zoZq, p2sf, 8bof, kZFF).
* Supported by extensive, well-motivated experiments (8bof, kZFF).
* Demonstrating clear benefits and consistent improvements, especially for minor classes (zoZq, p2sf).

**We will thoroughly incorporate their feedback into the camera-ready version.**

Please refer to **the attached PDF** for the updated figures and tables.

We hope our responses address all reviewers’ concerns and would greatly appreciate any further comments or clarifications.
***
# General response 1: Cost-benefit considerations of using LLMs (kZFF, 8bof)

A.  The motivation for using LLMs for synthetic tabular data generation stems from the limitations of existing methods, particularly in addressing underrepresented classes within imbalanced datasets. Traditional approaches often replicate existing biases, which can degrade ML classification performance, sometimes resulting in worse outcomes than using the original data. Our approach overcomes these by using in-context learning to present balanced, grouped examples to LLMs, effectively leveraging their advanced pattern recognition to mitigate data imbalance.

While the costs associated with LLMs are a valid concern, our findings show that these costs are manageable. For example, **generating 1,000 new samples costs less than $1 and takes under 100 seconds** with GPT API calls:
* Sick (27 features): $0.71, 85.0 seconds
* Income (14 features): $0.45, 95.6 seconds

Additionally, using open-source LLMs like Mistral or Llama2 requires approximately 25GB of GPU memory.

**Our approach also offers significant reductions in various overheads**:
* **Minimal preprocessing**: Unlike existing methods that often require extensive data preprocessing and handling of noisy or missing data, our LLM-based approach minimizes these steps.
* **No training**: Our method eliminates the need for extensive model training and hyperparameter tuning.
* **Single model versatility**: Our approach uses a single model across multiple datasets, unlike traditional methods that require separate models for each dataset.
* **Optimized prompting**: We provide a prompt specifically designed for synthetic tabular data generation, adaptable across various datasets with minimal adjustments, significantly reducing prompt engineering efforts.
* **Open access**: Our provided code allows users to easily generate new samples using open-source LLMs via Hugging Face or the GPT API.

**Despite the reduced costs, our method consistently outperforms existing baselines, producing the highest quality synthetic data**:
* **Consistent performance improvement**: To our knowledge, our method is the first to consistently outperform using only original data or SMOTE among the baselines, achieving state-of-the-art results. This was validated across six real-world datasets from diverse domains, including marketing, medical, finance, and social sciences (Tables 1, 6, 17, **A**).
* **Effective class imbalance mitigation**: Our method excels in generating balanced and representative data for minor classes in imbalanced datasets, preserving feature correlations (Figs. 4, 5) and improving overall model performance. As shown in **Fig. A**, our method achieves balanced improvements across key metrics, underscoring its robustness and practicality for real-world applications.

Our LLM-based approach offers a cost-effective and high-quality solution for synthetic tabular data generation, demonstrating the significant advantages of using LLMs for a wide range of real-world applications.
***
# General Response 2: Significance of performance improvements (kZFF, 8bof)

A. The relatively modest improvements in F1 score and balanced accuracy are due to the challenges of imbalanced datasets. As shown in **Fig. C**, while the gains in F1 score and balanced accuracy between 0 and 40K samples may seem modest, the sensitivity actually increases significantly from around 50% to 70%. This improvement balances performance across all classes, leading to a more meaningful outcome and greatly enhancing model usability.

Baselines often learn biases in the training data, resulting in abnormally high specificity at the expense of sensitivity. This imbalance can inflate balanced accuracy or F1 scores, but such performance is ineffective for real-world tasks. In contrast, our method significantly enhances sensitivity with only a slight reduction in specificity. As a result, **our method achieves the most balanced performance across all four metrics**, as shown in **Fig. A**.

Our method is the only approach among the 7 baselines that consistently improves both F1 score and balanced accuracy compared to using only the original data (Table 17). Extensive experiments across six real-world datasets from diverse domains (finance, medical, marketing, and social) using four classifiers, each tested five times, demonstrate the state-of-the-art performance of our method, with significant gains in the highly imbalanced Travel, Sick, and Income datasets (**Figs. B, D**). **While GReaT and TabDDPM are the closest baselines, they exhibit inconsistent performance.** As highlighted in **Table A**, they reduce original data performance in more than half of the cases (red). For example, both underperform on HELOC and Diabetes.

In stark contrast, **our method consistently outperforms the original data across all datasets** (blue). In the challenging Diabetes dataset, only our method improves over the original data. While specificity decreases in all methods, our method still achieves a superior balance of metrics, leading to greater practicality.

These results underscore that our method offers the greatest practical utility and a meaningful performance advantage among the models tested.

---

### Decision · Program_Chairs · 2024-09-25

**Decision:**

Accept (poster)

**Comment:**

The paper explores the effectiveness of LLMs in generating synthetic tabular data to address class imbalance in machine learning tasks. It introduces an approach that leverages in-context learning with LLMs to generate realistic and balanced synthetic data, enhancing the performance of machine learning models, particularly for minority classes in imbalanced datasets. The study investigates various prompt design strategies, including data formatting, class presentation, and variable mapping, finding that a CSV-style format and class-balanced grouping significantly improve data generation quality. The approach is validated across multiple real-world datasets, demonstrating its ability to improve classification performance and generate high-quality data without extensive model fine-tuning.

The paper is well-written and easy to follow. The reviewers had some questions and concerns about the experiments and analysis of the paper. During the rebuttal, the authors addressed these issues, and the reviewers were satisfied with the responses and additional experiments. Therefore, I recommend accepting the paper.